# Adversarial Robustness of Nonparametric Regression

**Parsa Moradi**[*]
University of Minnesota
`moradi@umn.edu`

**Hanzaleh Akbarinodehi**[*]
University of Minnesota
`akbar066@umn.edu`

**Mohammad Ali Maddah-Ali**
University of Minnesota
`maddah@umn.edu`

## Abstract

In this paper, we investigate the adversarial robustness of nonparametric regression, a fundamental problem in machine learning, under the setting where an adversary can arbitrarily corrupt a subset of the input data. While the robustness of parametric regression has been extensively studied, its nonparametric counterpart remains largely unexplored. We characterize the adversarial robustness in nonparametric regression, assuming the regression function belongs to the second-order Sobolev space (i.e., it is square integrable up to its second derivative).

The contribution of this paper is two-fold: (i) we establish a minimax lower bound on the estimation error, revealing a fundamental limit that no estimator can overcome, and (ii) we show that, perhaps surprisingly, the classical smoothing spline estimator, when properly regularized, exhibits robustness against adversarial corruption. These results imply that if $o(n)$ out of $n$ samples are corrupted, the estimation error of the smoothing spline vanishes as $n \to \infty$. On the other hand, when a constant fraction of the data is corrupted, no estimator can guarantee vanishing estimation error, implying the optimality of the smoothing spline in terms of maximum tolerable number of corrupted samples.

## 1 Introduction

In recent years, machine learning (ML) models have increasingly relied on data from diverse sources and are often deployed in distributed or decentralized computing environments [1–7]. These settings introduce new attack surfaces and create incentives for adversaries to corrupt data or disrupt learning algorithms [8–11]. This has motivated a growing body of research initiatives aimed at understanding and mitigating the impact of adversarial behavior [12–21].

One of the fundamental problems in ML is regression, which aims to estimate an unknown function $f$ based on observed noisy data [22]. This task is generally categorized into two approaches: *parametric* regression, which assumes $f$ is a parametric function with known structure, and *nonparametric* regression, which makes minimal assumptions on $f$, allowing it to belong to a wide class of functions such as Sobolev or Hölder spaces [23, 24]. Regression underpins many machine learning tasks, and understanding its robustness to adversarial corruption is a critical objective.

Adversarial robustness in parametric regression has been extensively studied [25–31]. Many approaches leverage tools from classical robust statistics [32, 33], adapting techniques like trimmed means, median-of-means, and $M$-estimators to modern high-dimensional settings [34–36]. These methods benefit from the structural constraints of parametric models, which narrow the hypothesis space and simplify the alleviation of adversarial attacks. In contrast, robustness in nonparametric regression is considerably more challenging due to the absence of such structure, which makes the models more vulnerable to adversarial attacks [37–40].

In this work, we address the problem of nonparametric regression under adversarial corruption. We consider a setting in which one observes $n$ pairs $\{(x_i, \widetilde{y}_i)\}_{i=1}^n$, where the responses $\widetilde{y}_i$ may be

---

[*]Equal contribution.

39th Conference on Neural Information Processing Systems (NeurIPS 2025).

partially corrupted by an adversary. Specifically, the adversary arbitrarily choose $\widetilde{y}_i$, for all $i \in \mathcal{A}$, where $\mathcal{A}$ is an unknown subset of $\{1, \ldots, n\}$ with cardinality at most $q < n$. For each $i \notin \mathcal{A}$, the observed response is $\widetilde{y}_i = f(x_i) + \varepsilon_i$, where $f \colon \Omega \to \mathbb{R}$ is the unknown regression function with domain $\Omega \subset \mathbb{R}$, and $\{\varepsilon_i\}_{i \in [n] \setminus \mathcal{A}}$ are i.i.d. noise variables with zero mean and variance at most $\sigma^2$.

In this paper, we assume that regression function $f$ belongs to the second-order Sobolev space, consisting of functions that are square-integrable up to the second derivative over $\Omega$. The objective of non-parametric regression is to produce $\hat{f}$ as an estimation of $f$ based on $\{(x_i, \widetilde{y}_i)\}_{i=1}^n$. To evaluate the performance of $\hat{f}$ in the presence of adversarial corruption, we use the following metrics [24]:

$$R_2(f, \hat{f}) := \mathbb{E}_{\boldsymbol{\varepsilon}} \left[ \sup_{\mathcal{S}} \|f - \hat{f}\|_{L_2(\Omega)}^2 \right], \qquad R_\infty(f, \hat{f}) := \mathbb{E}_{\boldsymbol{\varepsilon}} \left[ \sup_{\mathcal{S}} \|f - \hat{f}\|_{L_\infty(\Omega)}^2 \right],$$

where $\boldsymbol{\varepsilon} := [\epsilon_1, \ldots, \epsilon_n]$ and $\mathcal{S}$ denotes the adversarial strategy that can corrupt up to $q$ samples. Our goal is to characterize $\inf_{\hat{f}} R_2(f, \hat{f})$ and $\inf_{\hat{f}} R_\infty(f, \hat{f})$ over all estimators, assuming $f$ belongs to the second-order Sobolev space, under the setting where the adversary may corrupt up to $q$ samples.

The contributions of this paper are two-fold:

- **A Computationally-Efficient Estimator (Theorem 1):** We prove that the classical smoothing spline estimator retains robustness against adversarial corruption. This estimator selects $\hat{f}$ from the second-order Sobolev space, by minimizing the empirical error $\frac{1}{n} \sum_{i=1}^n (g(x_i) - \widetilde{y}_i)^2$, regularized by $\lambda \int \hat{f}''(x)^2 \, dx$, where $\lambda > 0$ controls the level of smoothness [41]. Smoothing splines are computationally efficient, with $\mathcal{O}(n)$ complexity of fitting and evaluating, leveraging B-spline basis functions [42, 43], and have found wide applications in statistics and machine learning [44–47]. Note that classical nonparametric methods are not necessarily adversarially robust. For instance, the Nadaraya–Watson (NW) estimator [48] can be fragile even under a small number of adversarial corruptions [39]. It is therefore surprising that a computationally efficient nonparametric regression method, such as smoothing splines, also exhibits adversarial robustness.

  While smoothing splines have been extensively studied in non-adversarial settings [41, 49–55], their robustness properties against adversarial corruption were not previously understood. In this paper, we show that if the adversary corrupts at most $q = o(n)$ samples and the regression function $f$ belongs to a second-order Sobolev space, then the smoothing spline estimator achieves $R_2 \to 0$ and $R_\infty \to 0$ as $n \to \infty$. This result further provides an upper bound on $\inf_{\hat{f}} R_2(f, \hat{f})$ and $\inf_{\hat{f}} R_\infty(f, \hat{f})$ as functions of $n$ and $q$.

- **Minimax Lower-Bound (Theorem 2):** We derive minimax lower bounds on $\inf_{\hat{f}} R_2(f, \hat{f})$ and $\inf_{\hat{f}} R_\infty(f, \hat{f})$, expressed as functions of $n$ and $q$. These bounds characterize the fundamental limits of estimation accuracy: no estimator can achieve better rates over the second-order Sobolev space under adversarial corruption.

  A key implication of this result is that when $q = \Theta(n)$, no estimator can achieve vanishing error as $n \to \infty$. This highlights that smoothing splines are not only computationally efficient but also optimal in terms of the maximum number of tolerable adversarial corruptions (see Corollary 4).

To better understand the results of this paper, we examine their implications in the regime of large $n$. In this regime, our results can be concisely summarized in Figure 1. This figure illustrates the rate of convergence, defined as $-\log_n R_2(f, \hat{f})$ or respectively, $-\log_n R_\infty(f, \hat{f})$, as a function of $q$, or equivalently, $\frac{\log q}{\log n} = \log_n q$, as $n \to \infty$. The red curves represent the impossibility result, indicating that no estimator can achieve a convergence rate beyond this bound for all functions in second-order Sobolev space (Theorem 2). The blue curve shows the convergence rate of the smoothing spline estimator in the presence of adversarial samples, as established in Theorem 1.

It is worth noting that, as shown in Figure 1a, when $\log_n q < \frac{2}{5}$, the smoothing spline achieves the minimax-optimal convergence rate for metric $R_2$ . For $\frac{2}{5} \leq \log_n(q) < 1$, in metric $R_2$ (Figure 1a), and for $0 \leq \log_n(q) < 1$ in metric $R_\infty$ (Figure 1b), while the smoothing splines offers vanishing estimation error for large $n$, the rate of convergence may not be optimum (there is a gap between the rate of convergence in smoothing splines and the minimax outer-bound). This theoretical results are supported with the simulation experiments results (see Section 4).

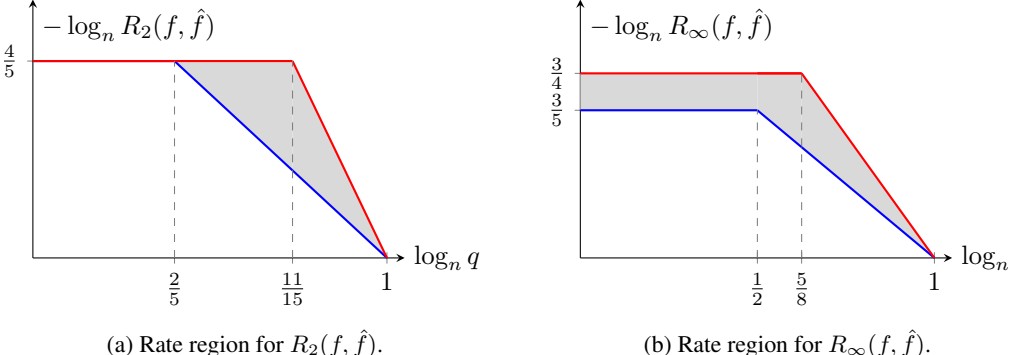

(a) Rate region for $R_2(f, \hat{f})$.

(b) Rate region for $R_\infty(f, \hat{f})$.

Figure 1: Rates of convergence for estimation error $R_2(f, \hat{f})$ and $R_\infty(f, \hat{f})$, as $n \to \infty$, and for any $f$ belongs second-order Sobolev space (for non-asymptotic analysis, see Theorems 1 and 2). The blue curves represent the minimum rate achieved by the smoothing spline estimator. The red curves denote minimax outer bounds that are impossible to beat. Specifically, for $q = o(n)$, for the smoothing spline estimator, both $R_2$ and $R_\infty$ converge to zero as $n \to \infty$. When $q = \Theta(n)$, we show that no estimator can achieve vanishing error, establishing a fundamental limit on robustness. This result highlights that smoothing splines are optimal in terms of the maximum tolerable number of adversarial corruptions (see Corollary 4).

This paper is organized as follows. Section 2 presents the problem formulation. Section 3 presents our main results by providing an upper and lower bounds under adversarial corruption. Section 4 provides simulation results, and Section 5 reviews related works.

**Notation.** Throughout the paper, we use $[n] := \{1, 2, \ldots, n\}$ and denote the cardinality of a set $\mathcal{A}$ by $|\mathcal{A}|$. Derivatives of scalar functions are written as $f'$, $f''$, and, more generally, $f^{(k)}$ for the $k$-th derivative. The quantities $\|g\|_{L^2(\Omega)}$ and $\|g\|_{L^\infty(\Omega)}$ denote the $L_2$-norm and the supremum norm of a function $g(\cdot)$ over $\Omega$. The space $\mathcal{W}^2(\Omega)$ refers to the second-order Sobolev space, consisting of square-integrable functions on $\Omega$ whose first and second derivatives are also square-integrable on $\Omega$. We write $a \lesssim b$ to indicate that there exists a constant $C > 0$ such that $a \leq Cb$, and similarly $a \gtrsim b$ to mean $a \geq Cb$ for some constant $C > 0$.

## 2 Problem Formulation

Let $f : [a, b] \to \mathbb{R}$ be in $\mathcal{W}^2([a, b])$. The objective is to estimate $f$, from observations at fixed design points $x_i \in (a, b)$ for $i \in [n]$. Instead of observing a noisy version of responses (as in standard regression problem [22]), we are given (possibly) adversarially corrupted outputs $\{\widetilde{y}_i\}_{i=1}^n$, defined as:

$$\widetilde{y}_i = \begin{cases} f(x_i) + \varepsilon_i, & \text{if } i \notin \mathcal{A}, \\ *, & \text{if } i \in \mathcal{A}, \end{cases}$$

where $\{\varepsilon_i\}_{i \in [n] \setminus \mathcal{A}}$ are i.i.d. noise variables with zero mean and variance at most $\sigma^2$, and $\mathcal{A} \subseteq [n]$ is an unknown subset of indices corresponding to adversarially corrupted observations. Here, $*$ denotes an arbitrary value chosen strategically by the adversary to mislead the estimator. We assume $|\mathcal{A}| \leq q$, for some known $q \in \mathbb{N}$.

Let $\Omega := [a, b]$ denote the domain of the design points, and $\boldsymbol{\varepsilon} = (\varepsilon_i)_{i \in [n] \setminus \mathcal{A}}$ denote the noise vector. Following [24], we evaluate the performance of any estimator $\hat{f}$ using two metrics, $R_2(f, \hat{f})$ and $R_\infty(\hat{f})$, where

$$R_2(f, \hat{f}) = \mathbb{E}_{\boldsymbol{\varepsilon}} \left[ \sup_{\mathcal{S}} \left\| f - \hat{f} \right\|_{L_2(\Omega)}^2 \right], \tag{1}$$

$$R_\infty(f, \hat{f}) = \mathbb{E}_{\boldsymbol{\varepsilon}} \left[ \sup_{\mathcal{S}} \left\| f - \hat{f} \right\|_{L_\infty(\Omega)}^2 \right], \tag{2}$$

where $\mathcal{S}$ denotes the strategy, chosen by the adversary, in choosing the subset $\mathcal{A}$ and the value of $\widetilde{y}_i$, for $i \in \mathcal{A}$, as long as $|\mathcal{A}| \leq q$. The supremum over $\mathcal{S}$ considers the worst-case adversarial attack, aiming to maximize estimation error for $\hat{f}$.

In this paper, the objective is to find $\hat{f}$, that minimizes $R_2(f, \hat{f})$ or $R_\infty(f, \hat{f})$, over all possible estimator functions $\hat{f}$, where $f$ is an arbitrary function in $\mathcal{W}^2(\Omega)$.

## 3  Main Results

In this section, we present our main results on the adversarial robustness of nonparametric regression. Without loss of generality, we assume that $\Omega = [0, 1]^2$. Let $\{x_i\}_{i=1}^n \subset \Omega$ denote the set of design points, and $f \in \mathcal{W}^2(\Omega)$ be the regression function.

First, we evaluate the robustness of the classical cubic smoothing spline estimator under adversarial corruption. In Theorem 1, we show that this estimator, as a computationally efficient [42, 43] and widely popular estimator [44–47], exhibits robustness to adversarial corruption.

The cubic smoothing spline estimator is defined as the solution to the following optimization problem:

$$\hat{f}_{\text{SS}}^a = \underset{g \in \mathcal{W}^2(\Omega)}{\arg\min} \left\{ \frac{1}{n} \sum_{i=1}^n \left( g(x_i) - \widetilde{y}_i \right)^2 + \lambda \int_\Omega \left( g''(x) \right)^2 dx \right\}. \tag{3}$$

Here, $\lambda > 0$ is a smoothing parameter that balances the fitness to the sample data, measured by $\frac{1}{n} \sum_{i=1}^n (g(x_i) - \widetilde{y}_i)^2$, and smoothness of the estimator, quantified by $\int_\Omega g''(x)^2 dx$.

We define the empirical distribution function $F_n$ associated with the design points $\{x_i\}_{i=1}^n$ as

$$F_n(x) = \frac{1}{n} \sum_{i=1}^n \mathbf{1}\{x_i \leq x\}, \tag{4}$$

where $\mathbf{1}\{x_i \leq x\}$ is the indicator function. We assume that $F_n$ converges uniformly to a continuously differentiable cumulative distribution function (CDF) $F$; that is,

$$\sup_{x \in \Omega} |F_n(x) - F(x)| \to 0 \quad \text{as} \quad n \to \infty.$$

This is a standard assumption in the related literature [24]. For the limiting CDF, i.e., $F(x)$, we assume that the density function $p(x) := F'(x)$ exists. In addition, similar to [39, 49], we assume that $p(x)$ is bounded away from zero, i.e., there exists a constant $p_{\min} > 0$ such that $\inf_{x \in \Omega} p(x) \geq p_{\min}$, and that $p(x)$ is three times continuously differentiable on $\Omega$.

We assume that the function $f$ is bounded; that is, for all $x \in \Omega$, $|f(x)| \leq m_1$ for some constant $m_1 \in \mathbb{R}$. Moreover, we assume that the adversary's corrupted values are also bounded, i.e., the adversary cannot inject arbitrarily large perturbations, satisfying $|\widetilde{y}_i| \leq m_2$ for $m_2 \in \mathbb{R}$ and $i \in \mathcal{A}$.

Finally, let $\Delta_{\max} := \sup_{x \in \Omega} \min_{i \in [n]} |x - x_i|$, $\Delta_{\min} := \min_{i \neq j} |x_i - x_j|$, denote the maximum gap from any point in $\Omega$ to the nearest design point, and the minimum separation between any two design points, respectively. Likewise to [50], we assume that their ratio is bounded by a constant, i.e., $\Delta_{\max}/\Delta_{\min} \leq k$, for some $k > 0$, ensuring that the design points are neither arbitrarily sparse nor overly clustered.

**Theorem 1** (Upper Bound). *Let $f \in \mathcal{W}^2(\Omega)$, and let $\hat{f}_{\text{SS}}^a$ denote the smoothing spline estimator defined in (3). Let $M = \max\{m_1, m_2\}$. Assume that $\lambda \to 0$ as $n \to \infty$ and $\lambda > n^{-2}$. Then, for sufficiently large $n$, we have*

$$R_2(f, \hat{f}_{\text{SS}}^a) \lesssim \lambda \int_\Omega \left( f''(x) \right)^2 dx + \frac{\sigma^2}{n\lambda^{1/4}} + \frac{q^2(M^2 + \sigma^2)}{n^2 \lambda^{1/2}}, \tag{5}$$

*and also*

$$R_\infty(f, \hat{f}_{\text{SS}}^a) \lesssim \lambda^{-1/4} \left( \lambda \int_\Omega \left( f''(x) \right)^2 dx + \frac{\sigma^2}{n\lambda^{1/4}} \right) + \frac{q^2(M^2 + \sigma^2)}{n^2 \lambda^{1/2}}. \tag{6}$$

---

[2]Note that any function $f \colon [a, b] \to \mathbb{R}$ can be transformed with scaling and shifting into a function $\tilde{f} \colon [0, 1] \to \mathbb{R}$ without affecting its Sobolev regularity or the scaling of the associated metrics.

For the proof details, see Appendix A. Here, we present a proof sketch: We first decompose each metric into two components using the triangle inequality. Specifically, we have

$$R_2(f, \hat{f}_{\mathrm{SS}}^a) \leq 2 \, \mathbb{E}_{\hat{\epsilon}} \left[ \sup_{\mathcal{S}} \left\| f - \hat{f}_{\mathrm{SS}} \right\|_{L_2(\Omega)}^2 \right] + 2 \, \mathbb{E}_{\hat{\epsilon}} \left[ \sup_{\mathcal{S}} \left\| \hat{f}_{\mathrm{SS}} - \hat{f}_{\mathrm{SS}}^a \right\|_{L_2(\Omega)}^2 \right], \tag{7}$$

$$R_\infty(f, \hat{f}_{\mathrm{SS}}^a) \leq 2 \, \mathbb{E}_{\hat{\epsilon}} \left[ \sup_{\mathcal{S}} \left\| f - \hat{f}_{\mathrm{SS}} \right\|_{L_\infty(\Omega)}^2 \right] + 2 \, \mathbb{E}_{\hat{\epsilon}} \left[ \sup_{\mathcal{S}} \left\| \hat{f}_{\mathrm{SS}} - \hat{f}_{\mathrm{SS}}^a \right\|_{L_\infty(\Omega)}^2 \right], \tag{8}$$

where $\hat{f}_{\mathrm{SS}}$ denotes the smoothing spline estimator fitted on clean (uncorrupted) data. More precisely, we have

$$\hat{f}_{\mathrm{SS}} := \arg \min_{g \in \mathcal{W}^2(\Omega)} \left\{ \frac{1}{n} \sum_{i=1}^n (y_i - g(x_i))^2 + \lambda \int_\Omega (g''(x))^2 \, dx \right\},$$

with $y_i = \tilde{y}_i$, for $i \in [n] \backslash \mathcal{A}$, and otherwise, for $i \in \mathcal{A}$, $y_i = f(x_i) + \epsilon_i$, for some i.i.d $\epsilon_i$. In addition, we have $\hat{\epsilon} = (\epsilon_i)_{i \in [n]}$.

The first term in each decompositions (7) and (8), quantifies the estimator's error in the absence of adversarial contamination, reflecting the classical estimation error. The second term, referred to as *adversarial deviation*, captures how much the adversarial estimator $\hat{f}_{\mathrm{SS}}^a$ deviates from its uncorrupted counterpart $\hat{f}_{\mathrm{SS}}$.

To bound the first term in decomposition (7), we rely on an established upper bounds for smoothing spline estimation [50, 51], which guarantee that, for sufficiently large $n$, we have

$$\mathbb{E}_{\hat{\epsilon}} \left[ \left\| f^{(j)} - \hat{f}_{\mathrm{SS}}^{(j)} \right\|_{L_2(\Omega)}^2 \right] \lesssim \lambda^{(2-j)/2} \int_\Omega (f''(x))^2 \, dx + \frac{\sigma^2}{n \lambda^{(2j+1)/4}}. \tag{9}$$

Applying (9) with $j = 0$ yields the desired bound for the first term in decomposition (7). For decomposition (8), the first term is bounded by combining (9) with $j = 0$ and $j = 1$, applying norm inequalities for Sobolev spaces [56], and using the Cauchy–Schwarz inequality, leading to

$$\mathbb{E}_{\hat{\epsilon}} \left[ \left\| f - \hat{f}_{\mathrm{SS}} \right\|_{L_\infty(\Omega)}^2 \right] \lesssim \lambda^{-1/4} \left( \lambda \int_\Omega (f''(x))^2 \, dx + \frac{\sigma^2}{n \lambda^{1/4}} \right). \tag{10}$$

To bound the second term in decompositions (7) and (8), we leverage the fact that the smoothing spline estimator is a linear smoother [41]. Specifically, the solution to (3) can be expressed in a kernel form as

$$\hat{f}_{\mathrm{SS}}^a(x) = \frac{1}{n} \sum_{i=1}^n W_n(x, x_i) \, \tilde{y}_i, \tag{11}$$

where $W_n(\cdot, \cdot)$ denotes the smoothing spline kernel (or weight function), which depends on the design points $\{x_i\}_{i \in [n]}$, sample size $n$, and the smoothing parameter $\lambda$. Using this representation, we have

$$\left| \hat{f}_{\mathrm{SS}}(x) - \hat{f}_{\mathrm{SS}}^a(x) \right| = \left| \frac{1}{n} \sum_{i=1}^n W_n(x, x_i) (y_i - \tilde{y}_i) \right| \overset{(a)}{=} \left| \frac{1}{n} \sum_{i \in \mathcal{A}} W_n(x, x_i) (y_i - \tilde{y}_i) \right|, \tag{12}$$

where $(a)$ follows from the fact that $y_i = \tilde{y}_i$, for $i \in [n] \backslash \mathcal{A}$. Using the Hölder inequality, and taking expectation, we show that

$$\mathbb{E}_{\hat{\epsilon}} \left[ \sup_{\mathcal{S}} \left\| \hat{f}_{\mathrm{SS}} - \hat{f}_{\mathrm{SS}}^a \right\|_{L_2(\Omega)}^2 \right] \lesssim \frac{q^2(M^2 + \sigma^2)}{n^2} \mathbb{E}_{\hat{\epsilon}} \left[ \sup_{\mathcal{S}, x \in \Omega, j \in [n]} |W_n(x, x_j)|^2 \right]. \tag{13}$$

Unfortunately, $W_n(\cdot, \cdot)$ does not admit an analytically tractable form [52, 53] for directly bounding its supremum in (13). However, a substantial body of research [52–55] has focused on approximating $W_n(\cdot, \cdot)$ with analytically tractable functions, known as *equivalent kernels*, denoted by $\widehat{W}_n(x, s)$. We leverage such approximations in our analysis to derive an upper bound for (13), leading to

$$\mathbb{E}_{\hat{\epsilon}} \left[ \sup_{\mathcal{S}} \left\| \hat{f}_{\mathrm{SS}}^a - \hat{f}_{\mathrm{SS}} \right\|_{L_2(\Omega)}^2 \right] \lesssim \frac{q^2(M^2 + \sigma^2)}{n^2 \lambda^{1/2}}. \tag{14}$$

We also take similar steps to derive

$$\mathbb{E}_{\hat{\epsilon}} \left[ \sup_{\mathcal{S}} \left\| \hat{f}_{\mathrm{SS}}^a - \hat{f}_{\mathrm{SS}} \right\|_{L_\infty(\Omega)}^2 \right] \lesssim \frac{q^2(M^2 + \sigma^2)}{n^2 \lambda^{1/2}}. \tag{15}$$

Combining (9), (10), (14), and (15) completes the proof of Theorem 1.

**Corollary 1** (Convergence Rate of $R_2(f, \hat{f})$). *Assume the conditions of Theorem 1 hold, and $q = \Theta(n^\beta)$ for some $\beta \in [0, 1]$. Then, by choosing $\lambda = \mathcal{O}(n^{-4/5})$ for $\beta \le 0.4$ and $\lambda = \mathcal{O}(n^{-4/3(1-\beta)})$ for $\beta > 0.4$, we have*

$$\inf_{\hat{f}} R_2(f, \hat{f}) \le R_2(f, \hat{f}_{\mathrm{SS}}^a) \le \begin{cases} \mathcal{O}\left(n^{-4/5}\right) & \text{for } \beta \le 0.4, \\ \mathcal{O}\left(n^{-4/3(1-\beta)}\right) & \text{for } \beta > 0.4, \end{cases} \tag{16}$$

*as depicted by the blue curve in Figure 1a.*

**Corollary 2** (Convergence Rate of $R_\infty(f, \hat{f})$). *Under the same assumptions as in Corollary 1, by choosing $\lambda = \mathcal{O}(n^{-4/5})$ for $\beta \le 0.5$ and $\lambda = \mathcal{O}(n^{-8/5(1-\beta)})$ for $\beta > 0.5$, we have*

$$\inf_{\hat{f}} R_\infty(f, \hat{f}) \le R_\infty(f, \hat{f}_{\mathrm{SS}}^a) \le \begin{cases} \mathcal{O}\left(n^{-3/5}\right) & \text{for } \beta \le 0.5, \\ \mathcal{O}\left(n^{-6/5(1-\beta)}\right) & \text{for } \beta > 0.5, \end{cases} \tag{17}$$

*as depicted by the blue curve in Figure 1b.*

Based on Corollaries 1 and 2, the thresholds at $\beta = 0.4$ for $R_2(f, \hat{f}_{\mathrm{SS}}^a)$ and $\beta = 0.5$ for $R_\infty(f, \hat{f}_{\mathrm{SS}}^a)$ indicate a phase transition: Below these points, the estimation error is dominated by noise, and adversarial corruption has no impact on the convergence rate. Beyond these thresholds, the adversary dictates the rate of convergence. In this scenario, we must choose a larger smoothing parameter $\lambda$ to smooth out the adversarial contribution in the data points.

Furthermore, for all $\beta \in [0, 1)$, the convergence rate of $R_\infty(f, \hat{f}_{\mathrm{SS}}^a)$ is slower than that of $R_2(f, \hat{f}_{\mathrm{SS}}^a)$, as established by Theorem 1. This reflects the greater sensitivity of $R_\infty$ estimation error to adversarial attacks: while metric $R_2$ averages the estimation error across the entire domain, metric $R_\infty$ is driven by the worst-case pointwise error, making it inherently more vulnerable to adversarial perturbations.

In the following theorem we provide a minimax lower bound for both metrics.

**Theorem 2.** *Let $P_\varepsilon$ denote the probability density function of the noise vector $\varepsilon$, with i.i.d zero-mean $\sigma^2$-variance entries, $f \in \mathcal{W}^2(\Omega)$ be the regression function. Then*

$$\inf_{\hat{f}} \sup_{f, \mathcal{S}, P_\varepsilon} R_2(f, \hat{f}) \gtrsim \left(\frac{q}{n}\right)^3 + \frac{1}{n^{4/5}}, \tag{18}$$

$$\inf_{\hat{f}} \sup_{f, \mathcal{S}, P_\varepsilon} R_\infty(f, \hat{f}) \gtrsim \left(\frac{q}{n}\right)^2 + \left(\frac{\log n}{n}\right)^{3/4}. \tag{19}$$

The full proof of Theorem 2 is provided in Appendix B. Here, we present a proof sketch. To establish Theorem 2, we reduce the minimax risk in (18) and (19) to a hypothesis testing problem [57]. Specifically, we construct two functions $f_1$ and $f_2$ in $\mathcal{W}^2(\Omega)$ with $L_2$ and $L_\infty$ distance, bounded away from zero (see Figure 2). However, given $n$ samples from either function, an adversary can corrupt up to $q$ of them, making it impossible for any estimator to reliably distinguish between $f_1$ and $f_2$. Consequently, no estimation approach can identify which function generated the data, and the average hypothesis testing error remains $1/2$. Applying [57, Proposition 5.1] yields the following lower bounds:

$$\inf_{\hat{f}} \sup_{f, \mathcal{S}, P_\varepsilon} R_2(f, \hat{f}) \gtrsim \left(\frac{q}{n}\right)^3, \tag{20}$$

$$\inf_{\hat{f}} \sup_{f, \mathcal{S}, P_\varepsilon} R_\infty(f, \hat{f}) \gtrsim \left(\frac{q}{n}\right)^2. \tag{21}$$

Furthermore, when $q = 0$, the adversarial model reduces to the classical non-adversarial setting, for which the minimax lower bounds have been established as $\inf_{\hat{f}} \sup_{f, P_\varepsilon} R_2(f, \hat{f}) \gtrsim n^{-4/5}$ and $\inf_{\hat{f}} \sup_{f, P_\varepsilon} R_\infty(f, \hat{f}) \gtrsim (\log n/n)^{3/4}$ [24]. Combining these with (20) and (21) completes the proof of Theorem 2.

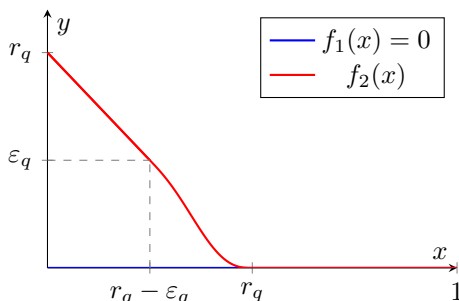

Figure 2: Construction of functions $f_1$ (blue) and $f_2$ (red) used in Theorem 2. Both functions belong to $\mathcal{W}^2([0, 1])$, where $f_1(x) = 0$ for all $x$, and $f_2(x)$ differs from $f_1$ only on the interval $[0, r_q]$, with $r_q = q/n$. The function $f_2$ is linear on $[0, r_q - \varepsilon_q]$, where $\varepsilon_q = r_q^2$, and transitions smoothly to zero on $[r_q - \varepsilon_q, r_q]$ via a degree-5 polynomial, ensuring $f_2 \in \mathcal{W}^2([0, 1])$. This construction induces a non-zero gap in both $L_2$ and $L_\infty$ norms, while enabling the adversary to obscure the difference by corrupting only $q$ samples, and making $f_1, f_2$ statistically indistinguishable. The details of this construction is provided in Appendix B.

**Corollary 3.** *Assuming $q = \Theta(n^\beta)$ for some $\beta \in [0, 1]$, we conclude from Theorem 2 that*

$$\inf_{\hat{f}} \sup_{f, \mathcal{S}, P_\varepsilon} R_2(f, \hat{f})) \geq \begin{cases} \mathcal{O}\left(n^{-4/5}\right) & \text{for } \beta \leq \frac{11}{15}, \\ \mathcal{O}\left(n^{-3(1-\beta)}\right) & \text{for } \beta > \frac{11}{15}, \end{cases} \tag{22}$$

$$\inf_{\hat{f}} \sup_{f, \mathcal{S}, P_\varepsilon} R_\infty(f, \hat{f}) \geq \begin{cases} \tilde{\mathcal{O}}\left(n^{-3/4}\right) & \text{for } \beta \leq \frac{5}{8}, \\ \tilde{\mathcal{O}}\left(n^{-2(1-\beta)}\right) & \text{for } \beta > \frac{5}{8}, \end{cases} \tag{23}$$

*as depicted by red curves in Figure 1.*

**Corollary 4** (On optimality of Smoothing Spline). *Assume the conditions of Theorem 1 hold and that $q = o(n)$. Then, by selecting the smoothing parameter $\lambda = (\frac{q}{n})^{4/3}$ when $q \geq n^{0.4}$ and $\lambda = n^{-0.8}$ when $q < n^{0.4}$, $R_2(f, \hat{f}_{SS}^a)$ vanishes as $n \to \infty$. Similarly, for $R_\infty(f, \hat{f}_{SS}^a)$, setting $\lambda = (\frac{q}{n})^{6/5}$ when $q \geq n^{0.5}$ and $\lambda = n^{-0.8}$ when $q < n^{0.5}$ ensures that $R_\infty(f, \hat{f}_{SS}^a)$ also converges to zero. Consequently, as long as $q = o(n)$, then $R_2(f, \hat{f}_{SS}^a)$ and $R_\infty(f, \hat{f}_{SS}^a)$ go to zero, as $n \to \infty$. Conversely, if $q = \Theta(n)$, according to Corollary 3, there exists a function $f \in \mathcal{W}^2(\Omega)$ such that, for any estimator $\hat{f}$, none of $R_2(f, \hat{f})$ and $R_\infty(f, \hat{f})$ converges to zero as $n \to \infty$. This implies that the classical cubic smoothing spline estimator is optimal with respect to maximum tolerable number of adversarial corruptions.*

## 4    Experimental Results

In this section, we present numerical experiments to validate the theoretical results. All experiments are conducted on a single CPU-only machine. The smoothing spline estimator is implemented using the `SciPy` package [58]. We consider two regression functions: (i) $f(x) = x \sin(x)$ over the domain $\Omega = [-10, 10]$ with $M = 100$, and (ii) a three-layer MLP network with weights initialized in $[-1, 1]$ and $M = 500$. The noise vector $\varepsilon$ is drawn independently from a Gaussian distribution with zero mean and variance $\sigma^2 = 1$.

To evaluate the adversarial robustness of the cubic smoothing spline estimator, we consider three distinct attack strategies:

- **Random Corruption Attack:** The adversary randomly selects $q$ out of the $n$ samples and replaces their response values with $M$.

- **Greedy Corruption Attack:** In this process, the attacker:
    1. Fits the baseline estimator on clean data.
    2. Computes $\ell_i = (\hat{f}(x_i) - y_i)^2$ for each sample.
    3. Identifies $i^\star = \arg\min_i \ell_i$.

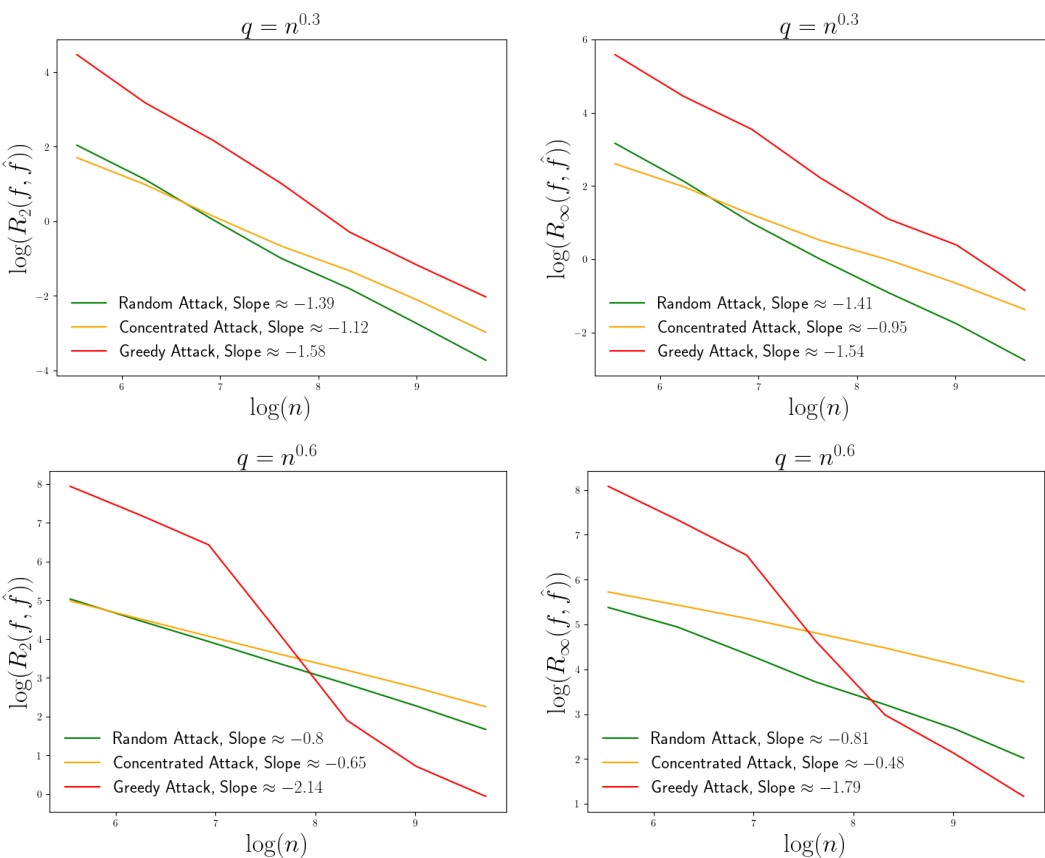

Figure 3: Log-log plots of error convergence rates for the cubic smoothing spline estimator $\hat{f} = \hat{f}_{\text{SS}}^a$ for $f(x) = x\sin(x)$ in the uniform design setting, where the input points converge to a uniform distribution. The top row shows $R_2(f, \hat{f})$ and $R_\infty(f, \hat{f})$ errors for $q = n^{0.3}$, along with the corresponding theoretical upper bounds of $\mathcal{O}(n^{-0.8})$ and $\mathcal{O}(n^{-0.6})$, respectively. The bottom row presents $R_2(f, \hat{f})$ and $R_\infty(f, \hat{f})$ errors for $q = n^{0.6}$, with theoretical upper bounds of $\mathcal{O}(n^{-0.53})$ and $\mathcal{O}(n^{-0.48})$, respectively.

4. Updates $y_{i^\star} \leftarrow y_{i^\star} + M \cdot \text{sign}(\hat{f}(x_{i^\star}) - y_{i^\star})$.

5. Repeats the process until $q$ points are corrupted.

- **Concentrated Corruption Attack:** The adversary targets $q$ consecutive samples centered around the median of the design points and modifies their corresponding labels to $M$.

For each attack strategy, we evaluate both $R_2(f, \hat{f}_{\text{SS}}^a)$ and $R_\infty(f, \hat{f}_{\text{SS}}^a)$ across a range of sample sizes $n$, and examine how these metrics scale with $n$ under varying levels of adversarial corruption. Additionally, for each experiment, we consider two settings for the design points: uniform and Gaussian. In the uniform and Gaussian settings, the design points $\{x_i\}_{i=1}^n$ converge to a uniform and a truncated Gaussian distribution over $\Omega$, respectively, as $n \to \infty$.

For the function $f(x) = x\sin(x)$, Figures 3 and 5 illustrate the behavior of $R_2(f, \hat{f}_{\text{SS}}^a)$ and $R_\infty(f, \hat{f}_{\text{SS}}^a)$ under uniform and Gaussian designs, respectively, for two corruption levels, $q = n^{0.3}$ and $q = n^{0.6}$. Similarly, for the MLP network, Figures 4 and 6 present the corresponding results.

As shown in these figures, the empirical convergence rates align well with the theoretical upper bounds established in Theorem 1. Specifically, Theorem 1 establishes that $R_2(f, \hat{f}_{\text{SS}}^a) \leq \mathcal{O}(n^{-0.8})$ and $R_\infty(f, \hat{f}_{\text{SS}}^a) \leq \mathcal{O}(n^{-0.6})$ for $q = n^{0.3}$, and $R_2(f, \hat{f}_{\text{SS}}^a) \leq \mathcal{O}(n^{-0.53})$ and $R_\infty(f, \hat{f}_{\text{SS}}^a) \leq \mathcal{O}(n^{-0.48})$ for $q = n^{0.6}$. These theoretical predictions align with the empirical convergence trends observed in

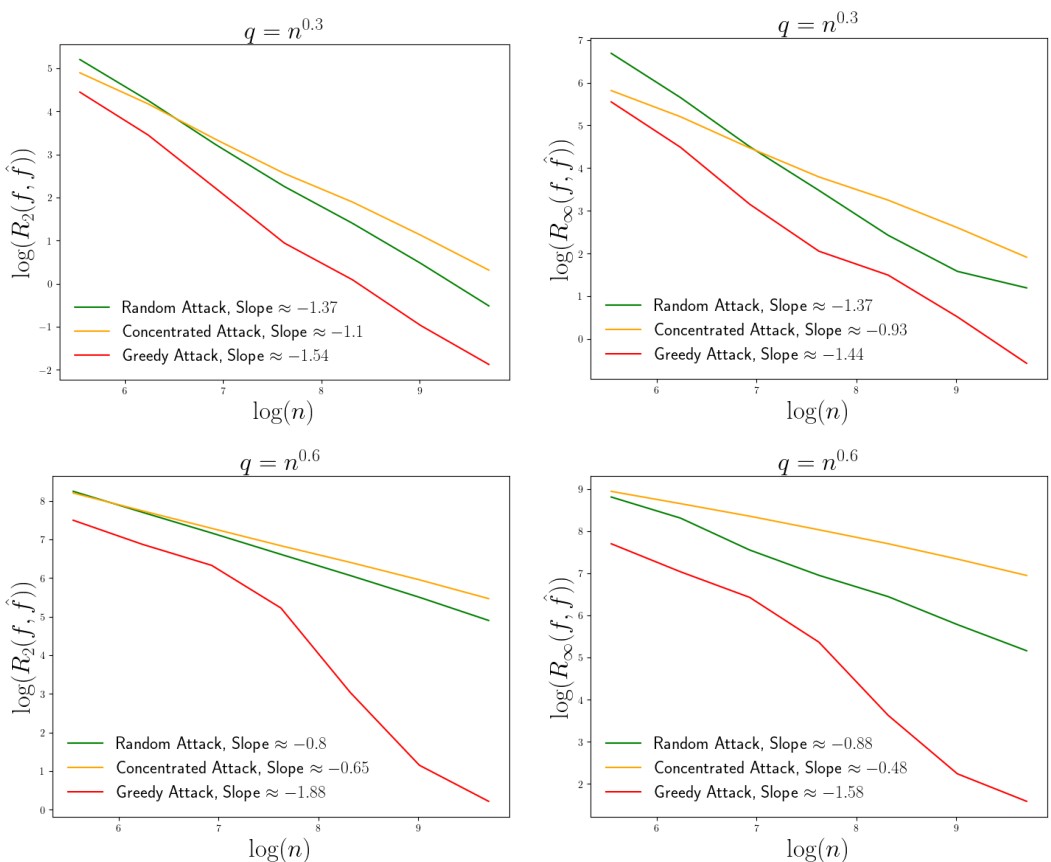

Figure 4: Log–log plots showing the convergence behavior of the cubic smoothing spline estimator $\hat{f} = \hat{f}^a_{\mathrm{SS}}$ when the ground-truth function is the MLP network, under the uniform design setting. The top row corresponds to the case $q = n^{0.3}$, with theoretical convergence rates of $\mathcal{O}(n^{-0.8})$ for $R_2(f, \hat{f})$ and $\mathcal{O}(n^{-0.6})$ for $R_\infty(f, \hat{f})$. The bottom row shows results for a higher corruption level, $q = n^{0.6}$, with respective theoretical upper bounds of $\mathcal{O}(n^{-0.53})$ and $\mathcal{O}(n^{-0.48})$.

the figures. Moreover, these figures show that the concentrated attack results in noticeably higher estimation error compared to the other two attack strategies.

It is important to note that these empirical rates are not expected to match the lower bounds from Theorem 2, since those bounds are minimax in nature. That is, they guarantee the existence of a worst-case function $f^\star \in \mathcal{W}^2(\Omega)$ for which no estimator can achieve faster convergence. Therefore, the lower bounds apply to such worst-case functions and not necessarily to all functions in $\mathcal{W}^2(\Omega)$, including the two functions in our experiments.

## 5 Related Work

Unlike parametric regression, which benefits from structural assumptions on the model class, nonparametric regression imposes minimal assumptions on the underlying function. This flexibility, makes it substantially more challenging to evaluate and guarantee adversarial robustness. Consequently, the literature on adversarial robustness in nonparametric settings remains relatively sparse. Nonetheless, several notable efforts have begun to address this gap.

As discussed earlier, the Nadaraya–Watson (NW) estimator is not robust to adversarial corruption and can fail even in the presence of a single corrupted sample [38, 39]. Classical robust estimation techniques, such as the Median-of-Means (MoM) estimator [59] and trimmed means [60], have been extended to nonparametric settings [37, 38, 61] to improve the robustness of the NW estimator. In the MoM approach, the data are partitioned into several groups, an NW estimator is fitted to each

group, and the median of the resulting estimates is taken. While this method enhances robustness to outliers, its performance degrades sharply when even a single corrupted sample appears in each group. Trimmed-mean methods, on the other hand, discard a fixed fraction of samples with extreme response values and fit the NW estimator on the remaining data. However, their effectiveness is limited when adversarial corruption is not uniformly distributed across the input space.

Zhao et al. [39] study adversarial robustness in kernel-based nonparametric regression by analyzing an $M$-estimator variant of the Nadaraya–Watson (NW) estimator [48, 62], deriving upper and minimax lower bounds for metrics based on $L_2$-norm and $L_\infty$-norm. In comparison to our setting, which assumes the regression function lies in a second-order Sobolev space, their work considers a first-order Hölder class. Their proposed estimator requires gradient descent with $\mathcal{O}(n \log(1/\epsilon))$ complexity to produce an estimation with precision $\epsilon$ on new data point. In contrast, the cubic smoothing spline accurately evaluates new data point in $\mathcal{O}(n)$ time, offering greater computational efficiency.

Several works also have studied the robustness of nonparametric classification. In [40], the authors analyze the robustness of nonparametric linear classifiers under arbitrary norms and mild regularity assumptions. The robustness of nearest neighbor classifiers against adversarial perturbations has been studied in [63] and a general attack framework applicable to a wide class of nonparametric classifiers is introduced in [64] and a data-pruning defense strategy to mitigate such attacks is proposed.

## 6  Conclusion

In this paper, we study the adversarial robustness of nonparametric regression when the underlying regression function belongs to $\mathcal{W}^2(\Omega)$. We prove that the cubic smoothing spline achieves vanishing $R_2$ and $R_\infty$ errors as long as the number of corrupted samples satisfies $q = o(n)$. We also establish lower bounds using a minimax argument. Notably, we show that cubic smoothing splines are optimal with respect to the maximum number of tolerable adversarial corruptions.

## Acknowledgment

This material is based upon work supported by the National Science Foundation under Grant CIF-2348638.

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

# A Proof of Theorem 1

In this section, we prove Theorem 1. Without loss of generality, we assume that $\Omega = [0,1]^3$. Recall that the solution of (3) is unique and the explicit formula for $\hat{f}_{\mathrm{SS}}^a$ is given by

$$\hat{f}_{\mathrm{SS}}^a(x) = \frac{1}{n}\sum_{i=1}^{n} W_n(x, x_i)\, \widetilde{y}_i, \tag{24}$$

where $W_n(x, x_i)$ denotes the smoothing spline weight function depending on $\{x_i\}_{i=1}^n$, the sample size $n$, and the smoothing parameter $\lambda$.

To facilitate the analysis, we define a second scenario in which the adversarial strategy is to be *honest*, that is, for any $i \in \mathcal{A}$, the adversary does not deviate from the clean data generation process and behaves as if it were non-adversarial. This allows us to construct a one-to-one correspondence between the realizations of the adversarial and honest scenarios such that for each $i \notin \mathcal{A}$, the observed responses $y_i$ are identical across both settings, while for $i \in \mathcal{A}$, the responses may differ: in Scenario 1 (adversarial), the adversary may introduce arbitrary deviations, whereas in Scenario 2 (honest), the responses follow the true underlying model.

In this second setting, we apply the same smoothing spline estimator to the uncorrupted data. The resulting estimator, which we denote by $\hat{f}_{\mathrm{SS}}$, is given by

$$\hat{f}_{\mathrm{SS}}(x) = \frac{1}{n}\sum_{i=1}^{n} W_n(x, x_i)\, y_i, \tag{25}$$

where $y_i$ denotes the uncorrupted response corresponding to input $x_i$, i.e., $y_i = \tilde{y}_i$, for $i \in [n]\backslash\mathcal{A}$, and otherwise, for $i \in \mathcal{A}$, $y_i = f(x_i) + \epsilon_i$, for some i.i.d $\epsilon_i$. We define $\hat{\epsilon} := (\epsilon_i)_{i\in[n]}$.

We now proceed to prove the bounds stated in Theorem 1. We first establish the upper bound for $R_2(f, \hat{f}_{\mathrm{SS}}^a)$ in (5), and subsequently turn to the bound for $R_\infty(f, \hat{f}_{\mathrm{SS}}^a)$ in (6).

By the definition of $R_2(f, \hat{f}_{\mathrm{SS}}^a)$, we have

$$R_2(f, \hat{f}_{\mathrm{SS}}^a) = \mathbb{E}_{\boldsymbol{\varepsilon}}\left[\sup_{\mathcal{S}}\left\|f - \hat{f}_{\mathrm{SS}}^a\right\|_{L_2(\Omega)}^2\right], \tag{26}$$

where $\boldsymbol{\varepsilon} = (\varepsilon_i)_{i\in[n]\backslash\mathcal{A}}$. First, observe that

$$\mathbb{E}_{\boldsymbol{\varepsilon}}\left[\sup_{\mathcal{S}}\left\|f - \hat{f}_{\mathrm{SS}}^a\right\|_{L_2(\Omega)}^2\right] = \mathbb{E}_{\hat{\boldsymbol{\epsilon}}}\left[\sup_{\mathcal{S}}\left\|f - \hat{f}_{\mathrm{SS}}^a\right\|_{L_2(\Omega)}^2\right],$$

which follows from the fact that $\left\|f - \hat{f}_{\mathrm{SS}}^a\right\|$ is independent of the noise terms $(\varepsilon_i)_{i\in\mathcal{A}}$. To proceed, we add and subtract $\hat{f}_{\mathrm{SS}}(x)$ inside the squared term:

$$\left(f(x) - \hat{f}_{\mathrm{SS}}^a(x)\right)^2 = \left(f(x) - \hat{f}_{\mathrm{SS}}(x) + \hat{f}_{\mathrm{SS}}(x) - \hat{f}_{\mathrm{SS}}^a(x)\right)^2. \tag{27}$$

Using AM-GM inequality, we obtain

$$\left(f(x) - \hat{f}_{\mathrm{SS}}^a(x)\right)^2 \leq 2\left(f(x) - \hat{f}_{\mathrm{SS}}(x)\right)^2 + 2\left(\hat{f}_{\mathrm{SS}}(x) - \hat{f}_{\mathrm{SS}}^a(x)\right)^2. \tag{28}$$

Substituting this bound into the definition of $R_2(f, \hat{f}_{\mathrm{SS}}^a)$, we get

$$R_2(f, \hat{f}_{\mathrm{SS}}^a) \leq 2\,\mathbb{E}_{\hat{\boldsymbol{\epsilon}}}\left[\sup_{\mathcal{S}}\left\|f - \hat{f}_{\mathrm{SS}}\right\|_{L_2(\Omega)}^2\right] + 2\,\mathbb{E}_{\hat{\boldsymbol{\epsilon}}}\left[\sup_{\mathcal{S}}\left\|\hat{f}_{\mathrm{SS}} - \hat{f}_{\mathrm{SS}}^a\right\|_{L_2(\Omega)}^2\right]. \tag{29}$$

To prove the upper bound in (5), it suffices to find appropriate bounds for the two terms appearing in (29). We begin by analyzing the first term involving the honest estimator $\hat{f}_{\mathrm{SS}}$:

$$\mathbb{E}_{\hat{\boldsymbol{\epsilon}}}\left[\sup_{\mathcal{S}}\left\|f - \hat{f}_{\mathrm{SS}}\right\|_{L_2(\Omega)}^2\right].$$

---

[3]Note that any function $f\colon [a,b] \to \mathbb{R}$ can be transformed with scaling and shifting into a function $\tilde{f}\colon [0,1] \to \mathbb{R}$ without affecting its Sobolev regularity or the scaling of the associated metrics.

To do so, we use the following theorem, which is a direct consequence of [50, Therorem 1.1], specialized to the second-order Sobolev space setting:

**Lemma 1.** *Let $I = [a, b] \subset \mathbb{R}$ be a bounded interval, and let the design points $\{x_i\}_{i=1}^n \subset I$ satisfy the quasi-uniformity condition*

$$\frac{\Delta_{\max}}{\Delta_{\min}} \leq k, \tag{30}$$

*for some constant $k > 0$, where*

$$\Delta_{\max} := \sup_{x \in I} \min_{i=1,\dots,n} |x - x_i|, \quad \Delta_{\min} := \min_{i \neq j} |x_i - x_j|. \tag{31}$$

*Then, for any $j = 0, 1, 2$, there exist constants $\lambda_0 > 0$, $P_0 > 0$, and $Q_0 > 0$, such that for all $n^{-4} \leq \lambda \leq \lambda_0$, we have*

$$\mathbb{E}_{\hat{\epsilon}}\left[\left\|f^{(j)} - \hat{f}_{\mathrm{SS}}^{(j)}\right\|_{L_2(I)}^2\right] \leq P_0\, \lambda^{\frac{2-j}{2}} \int_I \left(f''(x)\right)^2 dx + \frac{Q_0\, \sigma^2}{n\, \lambda^{\frac{2j+1}{4}}} \tag{32}$$

Here, $\hat{f}_{\mathrm{SS}}$ is the smoothing spline estimator applied to uncorrupted data, and $f^{(j)}$ denotes the $j$-th derivative of $f$. To bound the first term in (29), we invoke Lemma 1 with $j = 0$, corresponding to the $L_2(\Omega)$ error between the regression function $f$ and the honest smoothing spline estimator $\hat{f}_{\mathrm{SS}}$. This yields:

$$\mathbb{E}_{\hat{\epsilon}}\left[\left\|f - \hat{f}_{\mathrm{SS}}\right\|_{L_2(\Omega)}^2\right] \leq P_0\, \lambda \int_{\Omega} \left(f''(x)\right)^2\, dx + \frac{Q_0\, \sigma^2}{n\, \lambda^{1/4}}, \tag{33}$$

where $\lambda$ is the regularization parameter, and $P_0, Q_0 > 0$ are constants from Lemma 1.

To complete the proof of (5), we now seek to find an upper bound for the second term in (29), which captures the deviation between the adversarial and honest estimators:

$$\mathbb{E}_{\hat{\epsilon}}\left[\sup_{\mathcal{S}} \left\|\hat{f}_{\mathrm{SS}} - \hat{f}_{\mathrm{SS}}^a\right\|_{L_2(\Omega)}^2\right].$$

Note that from the kernel representations (24) and (25), we have

$$\hat{f}_{\mathrm{SS}}(x) - \hat{f}_{\mathrm{SS}}^a(x) = \frac{1}{n}\sum_{i=1}^n W_n(x, x_i)\,(y_i - \widetilde{y}_i). \tag{34}$$

Thus, for each $x \in \Omega$,

$$\left|\hat{f}_{\mathrm{SS}}(x) - \hat{f}_{\mathrm{SS}}^a(x)\right| = \left|\frac{1}{n}\sum_{i=1}^n W_n(x, x_i)\,(y_i - \widetilde{y}_i)\right|. \tag{35}$$

Note that for each $i$:

- If $i \notin \mathcal{A}$, there is no corruption, and $y_i = \widetilde{y}_i$.
- If $i \in \mathcal{A}$, the adversary may modify $y_i$, and since $f(x_i), \widetilde{y}_i \in [-M, M]$, we have

$$|y_i - \widetilde{y}_i| = |f(x_i) - \widetilde{y}_i + \epsilon_i| \leq |f(x_i) - \widetilde{y}_i| + |\epsilon_i| \leq 2M + |\epsilon_i|.$$

Thus, the sum above reduces to

$$\frac{1}{n}\sum_{i \in \mathcal{A}} W_n(x, x_i)\,(y_i - \widetilde{y}_i),$$

and we can bound

$$\left|\hat{f}_{\mathrm{SS}}(x) - \hat{f}_{\mathrm{SS}}^a(x)\right| \leq \frac{1}{n}\sum_{j \in \mathcal{A}} |W_n(x, x_j)|\,(2M + |\epsilon_i|) \leq \frac{1}{n}\sup_{x \in \Omega, j \in [n]} |W_n(x, x_j)| \cdot \sum_{j \in \mathcal{A}} (2M + \epsilon_i). \tag{36}$$

This implies that

$$\left\| \hat{f}_{\mathrm{SS}} - \hat{f}_{\mathrm{SS}}^a \right\|_{L_2(\Omega)}^2 \leq \left( \frac{\sup_{x\in\Omega, j\in[n]} |W_n(x,x_j)|}{n} \right)^2 \cdot \left( \sum_{i\in\mathcal{A}} (2M + |\epsilon_i|) \right)^2$$

$$\overset{(a)}{\leq} \left( \frac{\sup_{x\in\Omega, j\in[n]} |W_n(x,x_j)|}{n} \right)^2 \cdot \left( \sum_{i\in\mathcal{A}} 1^2 \right) \cdot \left( \sum_{i\in\mathcal{A}} (2M + |\epsilon_i|)^2 \right)$$

$$\overset{(b)}{\leq} \left( \frac{\sup_{x\in\Omega, j\in[n]} |W_n(x,x_j)|}{n} \right)^2 \cdot q \cdot \sum_{i\in\mathcal{A}} \left( 8M^2 + 2|\epsilon_i|^2 \right)$$

$$= \left( \frac{\sup_{x\in\Omega, j\in[n]} |W_n(x,x_j)|}{n} \right)^2 \cdot q \cdot \left( 8M^2 q + 2\sum_{i\in\mathcal{A}} \epsilon_i^2 \right), \tag{37}$$

where (a) and (b) follow from the Cauchy–Schwarz and AM–GM inequalities, respectively.

Taking expectations and supremum over $\mathcal{S}$ yields

$$\mathbb{E}_{\hat{\epsilon}} \left[ \sup_{\mathcal{S}} \left\| \hat{f}_{\mathrm{SS}} - \hat{f}_{\mathrm{SS}}^a \right\|_{L_2(\Omega)}^2 \right] \leq \frac{q^2 (8M^2 + 2\sigma^2)}{n^2} \sup_{x\in\Omega, j\in[n]} |W_n(x,x_j)|^2. \tag{38}$$

Now, to complete the proof of (5), it remains to find an upper bound for the kernel supremum term

$$\sup_{x,j\in[n]} |W_n(x,x_j)|.$$

Unfortunately, $W_n(\cdot,\cdot)$ does not admit an analytically tractable form [52, 53] for directly bounding its supremum in (13). However, a substantial body of research [52–55] has focused on approximating $W_n(\cdot,\cdot)$ with analytically tractable functions, known as *equivalent kernels*, denoted by $\widehat{W}_n(x,s)$. We leverage such approximations in our analysis to derive an upper bound.

Recall that we define the empirical distribution function $F_n$ as

$$F_n(x) = \frac{1}{n} \sum_{i=1}^n \mathbf{1}\{x_i \leq x\}. \tag{39}$$

We assume that the empirical distribution function $F_n$ converges to a cumulative distribution function $F$, i.e., $\alpha(n) := \sup_{x\in\Omega} |F_n(x) - F(x)|$ satisfies $\alpha(n) \longrightarrow 0$ as $n \to \infty$. Moreover, we assume that $F(x)$ is differentiable on $\Omega$ with density $p(x) = F'(x)$, and that there exists a constant $p_{\min} > 0$ such that

$$\inf_{x\in\Omega} p(x) \geq p_{\min}. \tag{40}$$

To proceed, according to [49], we define the equivalent kernel $\widehat{W}_n(x,s)$ as

$$\widehat{W}_n(x,s) = \frac{\lambda^{-1/4}}{2} (p(s)p(x))^{-3/8} e^{-\lambda^{-1/4}\varphi_0(x,s)} \sin\left( \lambda^{-1/4}\varphi_0(x,s) + \frac{\pi}{4} \right), \tag{41}$$

where the phase function $\varphi_0(x,s)$ is given by

$$\varphi_0(x,s) = 2^{-1/2} \int_{\min(x,s)}^{\max(x,s)} p(t)^{1/4} \, dt. \tag{42}$$

Based on [49, Theorem 1], for sufficiently large $n$, we have

$$\left| \widehat{W}_n(x,s) - W_n(x,s) \right| \leq C \left( \lambda^{-1/2}\alpha(n) + 1 \right), \tag{43}$$

where $C > 0$ is a constant independent of $n$, and the bound holds uniformly over all $x \in [0,1]$ and $s \in [\tau_1, \tau_2]$, where $0 < \tau_1 < \tau_2 < 1$.

Now note that

$$
\sup_{x \in \Omega, j \in [n]} |W_n(x, x_j)| = \sup_{x \in \Omega, j \in [n]} \left| \widehat{W}_n(x, x_j) + \left( W_n(x, x_j) - \widehat{W}_n(x, x_j) \right) \right| \tag{44}
$$

$$
\leq \sup_{x \in \Omega, j \in [n]} \left| \widehat{W}_n(x, x_j) \right| + \sup_{x \in \Omega, j \in [n]} \left| W_n(x, x_j) - \widehat{W}_n(x, x_j) \right|. \tag{45}
$$

Using the uniform approximation property established in (43), we can bound the second term:

$$
\sup_{x \in \Omega, j \in [n]} \left| W_n(x, x_j) - \widehat{W}_n(x, x_j) \right| \leq C \left( \lambda^{-1/2} \alpha(n) + 1 \right). \tag{46}
$$

Thus,

$$
\sup_{x \in \Omega, j \in [n]} |W_n(x, x_j)| \leq \sup_{x \in \Omega, j \in [n]} \left| \widehat{W}_n(x, x_j) \right| + C \left( \lambda^{-1/2} \alpha(n) + 1 \right)
$$

$$
\overset{(a)}{\leq} \frac{\lambda^{-1/4}}{2} (p_{\min})^{-3/4} + C \left( \lambda^{-1/2} \alpha(n) + 1 \right) \tag{47}
$$

where (a) follows from the definition of $\widehat{W}_n(x, x_j)$ in (41), and the fact that $\inf_{x \in \Omega} p(x) \geq p_{\min}$. Combining the decomposition in (29), the bound on the honest estimator error from (33), and the adversarial deviation bounds from (38) and (47), we obtain the final upper bound for $R_2(f, \hat{f}_{\mathrm{SS}}^a)$ stated in Theorem 1:

$$
R_2(f, \hat{f}_{\mathrm{SS}}^a) \leq 2P_0 \lambda \int_\Omega \left( f''(x) \right)^2 dx + \frac{2Q_0 \sigma^2}{n \lambda^{1/4}}
$$

$$
+ \frac{2q^2 (8M^2 + 2\sigma^2)}{n^2} \left[ \frac{\lambda^{-1/4}}{2} (p_{\min})^{-3/4} + C \left( \lambda^{-1/2} \alpha(n) + 1 \right) \right]^2. \tag{48}
$$

Therefore, in the regime where $\lambda \to 0$ as $n \to \infty$ and $\lambda > n^{-2} > n^{-4}$, there exist constants $E_1, E_2, E_3$ such that for sufficiently large $n$,

$$
R_2(f, \hat{f}_{\mathrm{SS}}^a) \leq E_1 \lambda \int_\Omega \left( f''(x) \right)^2 dx + \frac{E_2 \sigma^2}{n \lambda^{1/4}} + \frac{E_3 q^2 (M^2 + \sigma^2)}{n^2 \lambda^{1/2}} \left( 1 + \lambda^{-1/4} \alpha(n) + \lambda^{1/4} \right)^2. \tag{49}
$$

Since $\lambda^{1/4} \to 0$ as $n \to \infty$, the additive term $\lambda^{1/4}$ becomes negligible compared to 1 for sufficiently large $n$. Dropping this term and absorbing constants, we obtain

$$
R_2(f, \hat{f}_{\mathrm{SS}}^a) \lesssim \lambda \int_\Omega \left( f''(x) \right)^2 dx + \frac{\sigma^2}{n \lambda^{1/4}} + \frac{q^2 (M^2 + \sigma^2)}{n^2 \lambda^{1/2}} \left( 1 + \lambda^{-1/4} \alpha(n) \right)^2. \tag{50}
$$

For a continuous cumulative distribution function $F$, Serfling [65] shows that $\alpha(n) = n^{-1/2} \log \log n$ almost surely. Since $\lambda > n^{-2}$, it follows that $\lambda^{-1/4} \alpha(n) \to 0$ as $n \to \infty$. Therefore, for sufficiently large $n$, we have $1 + \lambda^{-1/4} \alpha(n) < 2$. As a result, we obtain

$$
R_2(f, \hat{f}_{\mathrm{SS}}^a) \lesssim \lambda \int_\Omega \left( f''(x) \right)^2 dx + \frac{\sigma^2}{n \lambda^{1/4}} + \frac{q^2 (M^2 + \sigma^2)}{n^2 \lambda^{1/2}}. \tag{51}
$$

This concludes the proof of the upper bound on $R_2(f, \hat{f}_{\mathrm{SS}}^a)$ in Theorem 1.

To complete the proof of Theorem 1, it remains to prove (6). To do so, we adopt a similar strategy as in the $L_2$ case, but adapted to the squared supremum norm. By the inequality $(a + b)^2 \leq 2a^2 + 2b^2$, we have

$$
\left\| f - \hat{f}_{\mathrm{SS}}^a \right\|_{L_\infty(\Omega)}^2 \leq 2 \left\| f - \hat{f}_{\mathrm{SS}} \right\|_{L_\infty(\Omega)}^2 + 2 \left\| \hat{f}_{\mathrm{SS}} - \hat{f}_{\mathrm{SS}}^a \right\|_{L_\infty(\Omega)}^2. \tag{52}
$$

Taking expectation and supremum over $\mathcal{S}$, we substitute into the definition of $R_\infty$ and obtain

$$R_\infty(f, \hat{f}_{\mathrm{SS}}^a) = \mathbb{E}_{\hat{\boldsymbol{\epsilon}}}\left[\sup_{\mathcal{S}}\left\|f - \hat{f}_{\mathrm{SS}}^a\right\|_{L_\infty(\Omega)}^2\right]$$

$$\leq 2\,\mathbb{E}_{\hat{\boldsymbol{\epsilon}}}\left[\sup_{\mathcal{S}}\left\|f - \hat{f}_{\mathrm{SS}}\right\|_{L_\infty(\Omega)}^2\right] + 2\,\mathbb{E}_{\hat{\boldsymbol{\epsilon}}}\left[\sup_{\mathcal{S}}\left\|\hat{f}_{\mathrm{SS}} - \hat{f}_{\mathrm{SS}}^a\right\|_{L_\infty(\Omega)}^2\right]. \tag{53}$$

From the pointwise bound established in (36), we have

$$\left|\hat{f}_{\mathrm{SS}}(x) - \hat{f}_{\mathrm{SS}}^a(x)\right| \leq \frac{2q(M + \max_i |\epsilon_i|)}{n} \sup_{x \in \Omega, j \in [n]} |W_n(x, x_j)|. \tag{54}$$

Applying the kernel estimate from (47), we conclude that

$$\left\|\hat{f}_{\mathrm{SS}} - \hat{f}_{\mathrm{SS}}^a\right\|_{L_\infty(\Omega)} \leq \frac{2q(M + \max_i |\epsilon_i|)}{n}\left(\frac{\lambda^{-1/4}}{2}(p_{\min})^{-3/4} + C\left(\lambda^{-1/2}\alpha(n) + 1\right)\right). \tag{55}$$

Squaring both sides and taking expectation and supremum over $\mathcal{S}$, we obtain

$$\mathbb{E}_{\hat{\boldsymbol{\epsilon}}}\left[\sup_{\mathcal{S}}\left\|\hat{f}_{\mathrm{SS}} - \hat{f}_{\mathrm{SS}}^a\right\|_{L_\infty(\Omega)}^2\right] \lesssim \frac{q^2(M^2 + \sigma^2)}{n^2\lambda^{1/2}}\left(1 + \lambda^{-1/4}\alpha(n) + \lambda^{1/4}\right)^2. \tag{56}$$

To complete the proof of (6), it remains to find an upper bound for the first term in (53), namely

$$\mathbb{E}_{\hat{\boldsymbol{\epsilon}}}\left[\sup_{\mathcal{S}}\left\|f - \hat{f}_{\mathrm{SS}}\right\|_{L_\infty(\Omega)}^2\right].$$

To do so, Since $f - \hat{f}_{SS} \in \mathcal{W}^2(\Omega)$, we can leverage Sobolev norms inequalities [56] and use the same arguments as in [66, Lemma 5] and obtain:

$$\left\|f - \hat{f}_{\mathrm{SS}}\right\|_{L_\infty(\Omega)}^2 \leq 2\left\|f - \hat{f}_{\mathrm{SS}}\right\|_{L_2(\Omega)} \cdot \left\|f' - \hat{f}_{\mathrm{SS}}'\right\|_{L_2(\Omega)} \tag{57}$$

Taking expectations on both sides of (57) and applying the Cauchy–Schwarz inequality, we obtain:

$$\mathbb{E}_{\hat{\boldsymbol{\epsilon}}}\left[\left\|f - \hat{f}_{\mathrm{SS}}\right\|_{L_\infty(\Omega)}^2\right] \leq 2\,\mathbb{E}_{\hat{\boldsymbol{\epsilon}}}\left[\left\|f - \hat{f}_{\mathrm{SS}}\right\|_{L_2(\Omega)} \cdot \left\|f' - \hat{f}_{\mathrm{SS}}'\right\|_{L_2(\Omega)}\right]$$

$$\leq 2\left(\mathbb{E}_{\hat{\boldsymbol{\epsilon}}}\left[\left\|f - \hat{f}_{\mathrm{SS}}\right\|_{L_2(\Omega)}^2\right]\right)^{1/2} \cdot \left(\mathbb{E}_{\hat{\boldsymbol{\epsilon}}}\left[\left\|f' - \hat{f}_{\mathrm{SS}}'\right\|_{L_2(\Omega)}^2\right]\right)^{1/2}. \tag{58}$$

Applying Lemma 1 with $j = 0$ and $j = 1$, we can bound the right-hand side using:

$$\mathbb{E}_{\hat{\boldsymbol{\epsilon}}}\left[\left\|f - \hat{f}_{\mathrm{SS}}\right\|_{L_2(\Omega)}^2\right] \leq P_0\lambda\int_\Omega (f''(x))^2\,dx + \frac{Q_0\sigma^2}{n\lambda^{1/4}}, \tag{59}$$

$$\mathbb{E}_{\hat{\boldsymbol{\epsilon}}}\left[\left\|f' - \hat{f}_{\mathrm{SS}}'\right\|_{L_2(\Omega)}^2\right] \leq P_0\lambda^{1/2}\int_\Omega (f''(x))^2\,dx + \frac{Q_0\sigma^2}{n\lambda^{3/4}}. \tag{60}$$

Substituting the bounds from (59) and (60) into (57), we obtain

$$\mathbb{E}_{\hat{\boldsymbol{\epsilon}}}\left[\left\|f - \hat{f}_{\mathrm{SS}}\right\|_{L_\infty(\Omega)}^2\right] \leq 2\left(P_0\lambda\int_\Omega (f''(x))^2\,dx + \frac{Q_0\sigma^2}{n\lambda^{1/4}}\right)^{1/2}$$

$$\times \left(P_0\lambda^{1/2}\int_\Omega (f''(x))^2\,dx + \frac{Q_0\sigma^2}{n\lambda^{3/4}}\right)^{1/2}. \tag{61}$$

Combining the decomposition in (53) with the bounds from (61) and (56), we obtain the following upper bound in the regime where $\lambda \to 0$ as $n \to \infty$ and $\lambda > n^{-2} \geq n^{-4}$:

$$R_\infty(f, \hat{f}_{\mathrm{SS}}^a) \lesssim \left(\lambda\int_\Omega (f''(x))^2\,dx + \frac{\sigma^2}{n\lambda^{1/4}}\right)^{1/2} \times \left(\lambda^{1/2}\int_\Omega (f''(x))^2\,dx + \frac{\sigma^2}{n\lambda^{3/4}}\right)^{1/2}$$

$$+ \frac{q^2(M^2 + \sigma^2)}{n^2\lambda^{1/2}}\left(1 + \lambda^{-1/4}\alpha(n) + \lambda^{1/4}\right)^2. \tag{62}$$

We now multiply and divide the first term by $\lambda^{1/4}$, yielding:

$$R_\infty(f, \hat{f}_{SS}^a) \lesssim \lambda^{-1/4} \left( \lambda \int_\Omega (f''(x))^2 \, dx + \frac{\sigma^2}{n\lambda^{1/4}} \right) + \frac{q^2(M^2 + \sigma^2)}{n^2\lambda^{1/2}} \left( 1 + \lambda^{-1/4}\alpha(n) + \lambda^{1/4} \right)^2.$$

By arguments similar to those used in the bound for $R_2(f, \hat{f}_{SS}^a)$, we can neglect both $\lambda^{1/4}$ and $\lambda^{-1/4}\alpha(n)$ compared to 1 for sufficiently large $n$. Thus, we obtain

$$R_\infty(f, \hat{f}_{SS}^a) \lesssim \lambda^{-1/4} \left( \lambda \int_\Omega (f''(x))^2 \, dx + \frac{\sigma^2}{n\lambda^{1/4}} \right) + \frac{q^2(M^2 + \sigma^2)}{n^2\lambda^{1/2}}. \tag{63}$$

This completes the proof of the upper bound on $R_\infty(f, \hat{f}_{SS}^a)$ in (6), and thereby concludes the proof of Theorem 1.

## B  Proof of Theorem 2

To prove Theorem 2, we first state and prove Lemma 2.

**Lemma 2.** *Let $P_1$ and $P_2$ denote two probability density functions of two distributions with common variance $\sigma^2 > 0$. Then, there exists $\alpha \in [0, 1]$, and two probability density functions $Q_1$ and $Q_2$ such that*

$$(1 - \alpha)P_1 + \alpha Q_1 = (1 - \alpha)P_2 + \alpha Q_2, \tag{64}$$

*where $Q_1$ and $Q_2$ are explicitly constructed from $P_1$ and $P_2$.*

*Proof.* Define $\alpha$ as:

$$\alpha = \frac{\int_{\{u:P_2(u) \geq P_1(u)\}} (P_2(u) - P_1(u)) \, du}{1 + \int_{\{u:P_2(u) \geq P_1(u)\}} (P_2(u) - P_1(u)) \, du} \leq 1. \tag{65}$$

Next, define $Q_1$ and $Q_2$ as:

$$Q_1(u) = \frac{1 - \alpha}{\alpha} (P_2(u) - P_1(u)) \mathbf{1}\{P_2(u) \geq P_1(u)\}, \tag{66}$$

$$Q_2(u) = \frac{1 - \alpha}{\alpha} (P_1(u) - P_2(u)) \mathbf{1}\{P_1(u) > P_2(u)\}, \tag{67}$$

where $\mathbf{1}\{\cdot\}$ denotes the indicator function.

By construction, both $Q_1(u)$ and $Q_2(u)$ are non-negative since the indicator functions restrict the support to regions where the corresponding differences are non-negative. We now show that $Q_1$ and $Q_2$ are valid probability density functions. Consider:

$$\int Q_1(u) \, du = \frac{1 - \alpha}{\alpha} \int (P_2(u) - P_1(u)) \mathbf{1}\{P_2(u) \geq P_1(u)\} \, du$$

$$= \frac{1 - \alpha}{\alpha} \int_{\{u:P_2(u) \geq P_1(u)\}} (P_2(u) - P_1(u)) \, du = 1. \tag{68}$$

By symmetry, the same argument shows that $\int Q_2(u) \, du = 1$ as well.

Hence, both $Q_1$ and $Q_2$ are valid densities. With this choice of $\alpha$, the following identity holds:

$$(1 - \alpha)P_1 + \alpha Q_1 = (1 - \alpha)P_2 + \alpha Q_2. \tag{69}$$

This completes the proof. □

We now prove Theorem 2, building on Lemma 2. We begin by establishing the lower bound for the metric $R_2$, as stated in (18); the proof for $R_\infty$, given in (19), follows by a similar argument. To do so, we reduce the minimax risk in (18) and (19) to a hypothesis testing problem [57]. Specifically, we construct two functions $f_1$ and $f_2$ in $\mathcal{W}^2(\Omega)$ with $L_2$ and $L_\infty$ distance, bounded away from zero (see Figure 2). However, given $n$ samples from either function, an adversary can corrupt up to $q$ of them, making it impossible for any estimator to reliably distinguish between $f_1$ and $f_2$. Consequently,

no estimation approach can identify which function generated the data, and the average hypothesis testing error remains $1/2$. Applying [57, Proposition 5.1] yields the lower bounds in Theorem 2. The details of the proof is as follows.

Throughout the proof, we assume a fixed design given by $x_i = i/n$ and $\varepsilon_i \sim \mathcal{N}(0, \sigma^2)$ are i.i.d noise samples drawn from a normal distribution with zero mean and variance $\sigma^2$, for $i \in [n]$.

Let $r_q = \frac{q}{n}$ and define $\varepsilon_q = r_q^2$. We construct two functions, $f_1$ and $f_2$, as follows. Set

$$f_1(x) = 0 \quad \text{for all } x \in [0, 1].$$

To define $f_2$, we construct a degree-5 polynomial $g(x)$ on the interval $[r_q - \varepsilon_q, r_q]$ that satisfies the following conditions:

$$g(r_q - \varepsilon_q) = \varepsilon_q, \tag{70}$$
$$g'(r_q - \varepsilon_q) = -1, \tag{71}$$
$$g''(r_q - \varepsilon_q) = 0, \tag{72}$$
$$g(r_q) = 0, \tag{73}$$
$$g'(r_q) = 0, \tag{74}$$
$$g''(r_q) = 0. \tag{75}$$

These six conditions uniquely determine a polynomial of degree 5, since there are six coefficients to solve for. Hence, such a polynomial $g$ exists and can be explicitly constructed. Now, define $f_2$ on the interval $[0, 1]$ by

$$f_2(x) = \begin{cases} r_q - x, & \text{if } x \in [0, r_q - \varepsilon_q], \\ g(x), & \text{if } x \in [r_q - \varepsilon_q, r_q], \\ 0, & \text{if } x > r_q. \end{cases}$$

It is straightforward to verify that $f_2 \in \mathcal{W}^2([0, 1])$, since both $f_2$ and its first and second derivatives have bounded norms over $\Omega$ (See Figure 2).

Note that $f_1$ and $f_2$ are close but not identical; their differences are concentrated on the interval $[0, r_q]$, and will be used to construct the lower bound.

For each sample $x_i$, the adversary proceeds as follows:

- If $x_i \geq r_q$, then $f_1(x_i) = f_2(x_i)$, so no corruption is needed: both models produce identical distributions for $\widetilde{y}_i$.

- If $x_i < r_q$, then $f_1(x_i) \neq f_2(x_i)$, and the adversary applies Lemma 2 to the pair of normal distributions

$$P_1^{(i)} := \mathcal{N}(f_1(x_i), \sigma^2), \qquad P_2^{(i)} := \mathcal{N}(f_2(x_i), \sigma^2),$$

obtaining a scalar $\alpha_i \in [0, 1]$ and auxiliary distributions $Q_1^{(i)}$ and $Q_2^{(i)}$ such that

$$(1 - \alpha_i)P_1^{(i)} + \alpha_i Q_1^{(i)} = (1 - \alpha_i)P_2^{(i)} + \alpha_i Q_2^{(i)}.$$

For each such $i$, the adversary acts:

- With probability $1 - \alpha_i$, leave $y_i$ uncorrupted (i.e., drawn from $P_1^{(i)}$ if $f = f_1$, or from $P_2^{(i)}$ if $f = f_2$).

- With probability $\alpha_i$, the adversary replaces $y_i$ by a draw from $Q_1^{(i)}$ if the true function is $f_1$, and from $Q_2^{(i)}$ if the true function is $f_2$.

For the above adversarial strategy, we have $|\mathcal{A}| \leq r_q n = q$. In addition, note that under model $f_1$, conditionally on $x_i$, the corrupted response $\widetilde{y}_i$ is distributed according to $(1 - \alpha_i)P_1^{(i)} + \alpha_i Q_1^{(i)}$, and under model $f_2$, it is distributed according to $(1 - \alpha_i)P_2^{(i)} + \alpha_i Q_2^{(i)}$. By construction of $Q_1^{(i)}$ and $Q_2^{(i)}$ in Lemma 2, these two mixtures are identical for each $i$.

Therefore, after adversarial corruption, the distribution of all observed data $\{\widetilde{y}_i\}_{i=1}^n$ is identical under $f_1$ and $f_2$. More precisely:

- For all $i$ with $x_i > r_q$, we have $f_1(x_i) = f_2(x_i)$, and hence $P_1^{(i)} = P_2^{(i)}$; no corruption is needed, and the distribution of $\widetilde{y}_i$ is the same under both models.
- For all $i$ with $x_i \leq r_q$, the adversary modifies the responses exactly so that the overall conditional distribution of $\widetilde{y}_i$ is matched across the two models.

Note that the constructed functions $f_1$ and $f_2$ are not identical: by definition, their difference measured by the metrics introduced in (1) and (2) is nonzero. However, the adversarial corruption strategy described above renders the corrupted data distribution identical under both $f_1$ and $f_2$. Consequently, no estimator can achieve better performance than random guessing between the two hypotheses. As a result, the minimax error under adversarial corruption remains bounded away from zero, establishing a nontrivial lower bound.

To prove (18), by starting from the definition of $R_2(f, \hat{f})$, we have

$$R_2(f, \hat{f}) = \mathbb{E}_\varepsilon \left[ \sup_{\mathcal{S}} \int_0^1 \left( f(x) - \hat{f}(x) \right)^2 dx \right], \tag{76}$$

where the expectation is over the noise $\varepsilon$, and the supremum is taken over all admissible adversarial strategies $\mathcal{S}$. Since Theorem 2 considers the worst-case function $f$, we obtain

$$\inf_{\hat{f}} \sup_{f \in \mathcal{W}^2(\Omega), \mathcal{S}, P_\varepsilon} R_2(f, \hat{f}) \geq \inf_{\hat{f}} \sup_{f \in \{f_1, f_2\}} \mathbb{E}_\varepsilon \left[ \int_0^1 \left( f(x) - \hat{f}(x) \right)^2 dx \right]. \tag{77}$$

As established earlier, the adversary makes the corrupted data distribution identical under both $f_1$ and $f_2$. Formally, let $\mathbb{P}_{f_1}^{(\mathcal{A})}$ and $\mathbb{P}_{f_2}^{(\mathcal{A})}$ denote the distributions over the corrupted datasets when the ground truth is $f_1$ or $f_2$, respectively. Thus, we have:

$$\mathbb{P}_{f_1}^{(\mathcal{A})} = \mathbb{P}_{f_2}^{(\mathcal{A})}.$$

That is, the total variation distance satisfies:

$$\mathrm{TV}(\mathbb{P}_{f_1}^{(\mathcal{A})}, \mathbb{P}_{f_2}^{(\mathcal{A})}) = 0. \tag{78}$$

This guarantees that no estimator can distinguish between them better than random guessing. To formalize this, we use Le Cam's two-point method [67, 68] (the hypothesis testing between two points), which states that for any estimator $\hat{f}$ and any pair $f_1, f_2$,

$$\inf_{\hat{f}} \sup_{f \in \{f_1, f_2\}} \mathbb{E}_\varepsilon \left[ \|\hat{f} - f\|_{L^2(\Omega)}^2 \right] \geq \frac{\|f_1 - f_2\|_{L^2(\Omega)}^2}{4} \cdot \left( 1 - \mathrm{TV}(\mathbb{P}_{f_1}^{(\mathcal{A})}, \mathbb{P}_{f_2}^{(\mathcal{A})}) \right).$$

Using (78), we obtain the following lower bound:

$$\inf_{\hat{f}} \sup_{f \in \{f_1, f_2\}} \mathbb{E} \left[ \|\hat{f} - f\|_{L^2(\Omega)}^2 \right] \geq \frac{1}{4} \|f_1 - f_2\|_{L^2(\Omega)}^2.$$

Consequently, following (77) we have

$$\inf_{\hat{f}} \sup_{f \in \mathcal{W}^2(\Omega), \mathcal{S}, P_\varepsilon} R_2(f, \hat{f}) \geq \frac{1}{4} \int_0^1 (f_1(x) - f_2(x))^2 dx. \tag{79}$$

Recall that $f_1(x) = 0$, and

$$f_2(x) = \begin{cases} r_q - x, & x \in [0, r_q - \varepsilon_q], \\ g(x), & x \in [r_q - \varepsilon_q, r_q], \\ 0, & x > r_q, \end{cases}$$

where $g(x)$ is a degree-5 polynomial satisfying the smoothness and boundary conditions described earlier. Therefore,

$$\int_0^1 (f_1(x) - f_2(x))^2 dx = \int_0^{r_q} f_2(x)^2 dx = \int_0^{r_q - \varepsilon_q} (r_q - x)^2 dx + \int_{r_q - \varepsilon_q}^{r_q} g(x)^2 dx$$

$$\geq \int_0^{r_q - \varepsilon_q} (r_q - x)^2 dx. \tag{80}$$

Note that since $\varepsilon_q = r_q^2$, we have

$$\int_0^{r_q - \varepsilon_q} (r_q - x)^2 dx = \int_{\varepsilon_q}^{r_q} u^2 du = \frac{r_q^3 - \varepsilon_q^3}{3} \gtrsim r_q^3 = \left(\frac{q}{n}\right)^3. \tag{81}$$

Therefore, we have

$$\inf_{\hat{f}} \sup_{f \in \mathcal{W}^2(\Omega), \mathcal{S}, P_\varepsilon} R_2(f, \hat{f}) \gtrsim r_q^3 = \left(\frac{q}{n}\right)^3. \tag{82}$$

Moreover, even in the absence of adversarial corruption (i.e., $q = 0$), it is well known from classical minimax theory in nonparametric regression [24] that

$$\inf_{\hat{f}} \sup_{f \in \mathcal{W}^2(\Omega)} \mathbb{E}\left[\left\|f - \hat{f}\right\|_{L_2(\Omega)}^2\right] \gtrsim n^{-4/5}. \tag{83}$$

Combining the two regimes, we obtain the following lower bound on the adversarial error:

$$\inf_{\hat{f}} \sup_{f \in \mathcal{W}^2(\Omega), \mathcal{S}, P_\varepsilon} R_2(f, \hat{f}) \gtrsim \left(\frac{q}{n}\right)^3 + n^{-4/5}. \tag{84}$$

This completes the proof of (18). To complete the proof of Theorem 2, we now establish a lower bound for $R_\infty$. Recall that

$$R_\infty(f, \hat{f}) = \mathbb{E}_\varepsilon\left[\sup_{\mathcal{S}} \left\|f - \hat{f}\right\|_{L_\infty(\Omega)}^2\right], \tag{85}$$

where the expectation is taken over the noise $\varepsilon$, and the supremum is over all adversarial corruption strategies $\mathcal{S}$. The norm $\|\cdot\|_{L_\infty(\Omega)}$ denotes the supremum norm over the interval $[0, 1]$.

As in the case of $R_2$, the adversary can construct corrupted data distributions under $f_1$ and $f_2$ that are indistinguishable. Consequently, no estimator can distinguish between the two hypotheses better than random guessing. Applying Le Cam's two-point method [67, 68] to the $L_\infty$ loss, we obtain:

$$\inf_{\hat{f}} \sup_{f \in \{f_1, f_2\}} \mathbb{E}_\varepsilon\left[\|\hat{f} - f\|_{L_\infty(\Omega)}^2\right] \geq \frac{\|f_1 - f_2\|_{L_\infty(\Omega)}^2}{4} \cdot \left(1 - \mathrm{TV}(\mathbb{P}_{f_1}^{(\mathcal{A})}, \mathbb{P}_{f_2}^{(\mathcal{A})})\right).$$

Therefore, we have:

$$\inf_{\hat{f}} \sup_{f \in \mathcal{W}^2(\Omega), \mathcal{S}, P_\varepsilon} R_\infty(f, \hat{f}) \geq \frac{\|f_1 - f_2\|_{L_\infty(\Omega)}^2}{4}. \tag{86}$$

Since $f_1(x) = 0$, we have $\|f_1 - f_2\|_{L_\infty(\Omega)} = \|f_2\|_{L_\infty(\Omega)} \geq f_2(0)$. From the definition of $f_2$, we have $f_2(0) = r_q$. Therefore,

$$\inf_{\hat{f}} \sup_{f \in \mathcal{W}^2(\Omega), \mathcal{S}, P_\varepsilon} R_\infty(f, \hat{f}) \gtrsim r_q^2 = \left(\frac{q}{n}\right)^2. \tag{87}$$

Moreover, in the absence of adversarial corruption (i.e., $q = 0$), the standard minimax rate for estimation under the supremum norm is known to satisfy (see [24])

$$\inf_{\hat{f}} \sup_{f \in \mathcal{W}^2(\Omega)} \mathbb{E}\left[\left\|f - \hat{f}\right\|_{L_\infty(\Omega)}^2\right] \gtrsim \left(\frac{\log n}{n}\right)^{3/4}. \tag{88}$$

Combining both contributions, we conclude that

$$\inf_{\hat{f}} \sup_{f \in \mathcal{W}^2(\Omega), \mathcal{S}, P_\varepsilon} R_\infty(f, \hat{f}) \gtrsim \left(\frac{q}{n}\right)^2 + \left(\frac{\log n}{n}\right)^{3/4}. \tag{89}$$

This completes the proof of (19), and thereby the proof of Theorem 2.

## C    Gaussian Setting Experiments

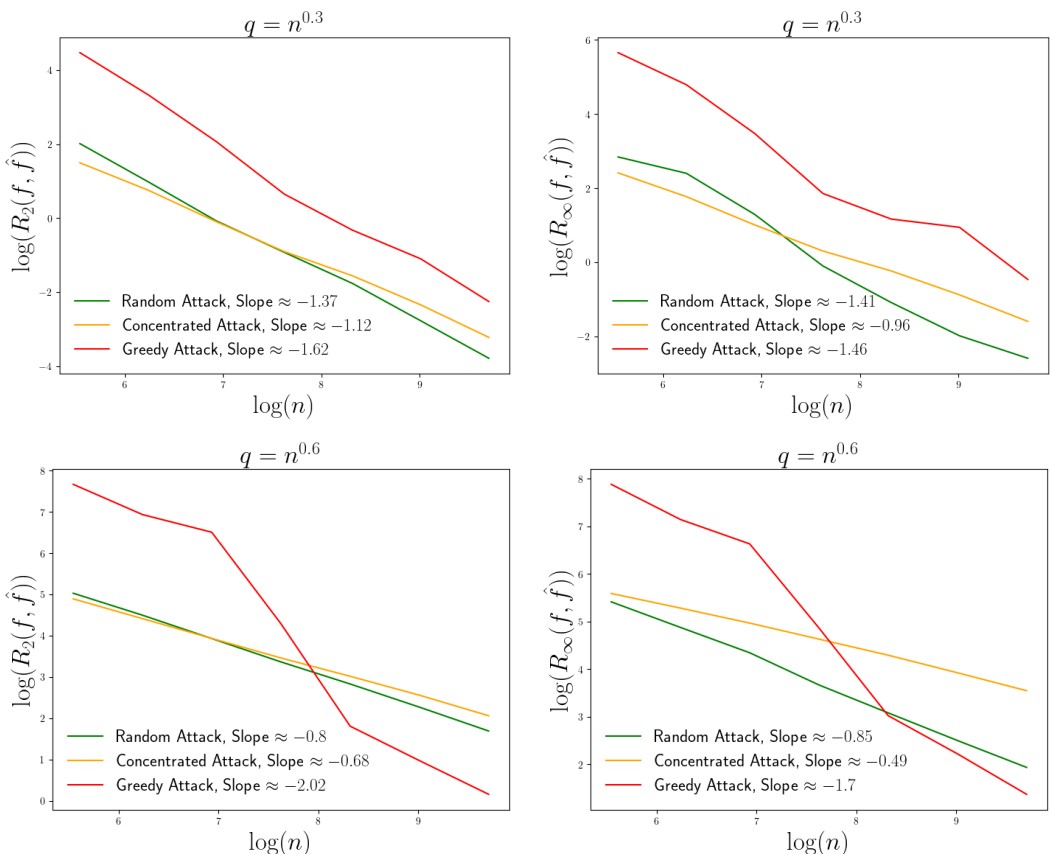

Figure 5: Log-log plots showing the convergence rate of the cubic smoothing spline estimator $\hat{f} = \hat{f}_{\mathrm{SS}}^{a}$ for $f(x) = x\sin(x)$ under a Gaussian design. The top row plots are results for $q = n^{0.3}$, with theoretical rates of $\mathcal{O}(n^{-0.8})$ for $R_2(f, \hat{f})$ and $\mathcal{O}(n^{-0.6})$ for $R_\infty(f, \hat{f})$. The bottom row corresponds to a higher corruption level, $q = n^{0.6}$, with respective theoretical upper bounds of $\mathcal{O}(n^{-0.53})$ and $\mathcal{O}(n^{-0.48})$.

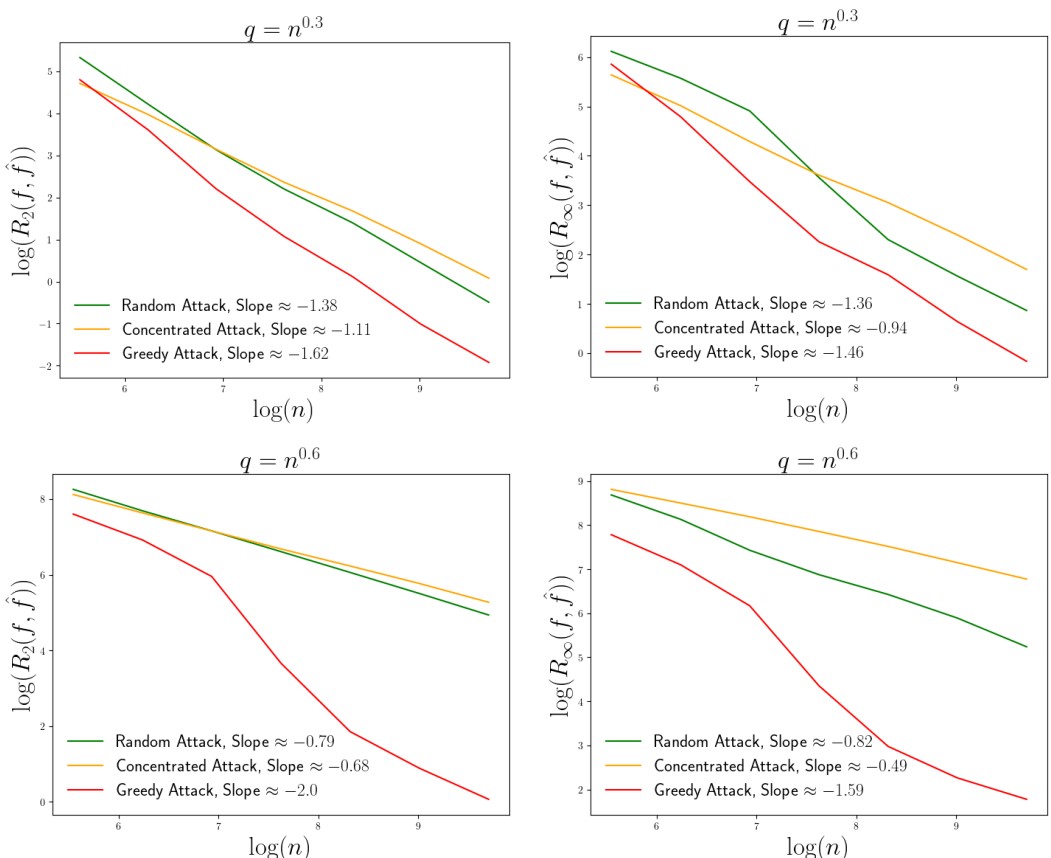

Figure 6: Log–log plots showing the convergence behavior of the cubic smoothing spline estimator $\hat{f} = \hat{f}_{\mathrm{SS}}^a$ when the ground-truth function is an MLP, under the Gaussian design. The top row corresponds to the case $q = n^{0.3}$, with theoretical convergence rates of $\mathcal{O}(n^{-0.8})$ for $R_2(f, \hat{f})$ and $\mathcal{O}(n^{-0.6})$ for $R_\infty(f, \hat{f})$. The bottom row shows results for a higher corruption level, $q = n^{0.6}$, with respective theoretical upper bounds of $\mathcal{O}(n^{-0.53})$ and $\mathcal{O}(n^{-0.48})$.

