# OpenReview forum: "Adversarial Robustness of Nonparametric Regression"
_NeurIPS.cc/2025/Conference — NeurIPS 2025 poster_

### Official Review · Reviewer_pg9c · 2025-06-05

**Clarity:** 3
**Significance:** 2
**Originality:** 3
**Rating:** 5
**Confidence:** 4

**Summary:**

This paper positions itself as one of the first to analyze the robustness properties of nonparametric regression models when the underlying function belongs to a second-order Sobolev space. It begins by deriving an upper bound on the risk of the smoothing spline estimator in the asymptotic setting. The authors then characterize the minimax optimal risk and discuss where and how the two bounds coincide.

**Questions:**

1. One high-level question that a reader may ask is why we should choose a nonparametric regression function over a parametric one (including large-scale deep networks) to be concerned about their vulnerability to adversarial attacks.

2. While the authors refer to the problem as the adversarial robustness of regression models, I do not see a significant difference between their setup and the standard robust regression framework with outliers—that is, $y=f(x)+e$ where $e$ follows a contamination model of the form $(1-\epsilon) N(0,\sigma^2)+\epsilon h$ where $h$ is an unown distribution. Given that, I analyze the problem from the perspective of robust estimation. What is the breakdown point of this estimator? For example, we know that the breakdown point of the median estimator is 50\%, meaning it can handle up to 50\% of the data being arbitrary without the estimator diverging. While I understand that the authors’ assumption about  range of $y \in [-M,M]$ prevents divergence, a more balanced basis for comparison would be beneficial.

3. I do not understand why the authors find it surprising that a computationally efficient nonparametric regression method, such as smoothing splines, demonstrates adversarial robustness. Given the objective function for spline fitting, which includes a penalty term that regularizes the function by penalizing roughness through the integrated squared second derivative, it is natural that smoother functions are less vulnerable to adversarial perturbations. Hence, adding a deviated sample is unlikely to cause a significant change in the fitted values.

4. What does q = $\boldsymbol{\Theta}(n^\beta)$ mean?

5. The authors discuss the asymptotic behavior of the estimator as $n \rightarrow \infty$, but they do not address the small-sample regime, which is often more relevant in practical regression settings. Do you have any intuition on how the guarantees might change in such cases?

6. In Figure 3, please include the curve for the best possible estimator in the same plot to clearly illustrate the gaps and assess how tight the results are.

7. In the experiments, the attack types are rather simple. Please consider using a greedy search to optimize the attack strategy, which will help evaluate how robust and tight the bounds truly are.

**Ethical Concerns:**

["NO or VERY MINOR ethics concerns only"]

**Final Justification:**

I had a detailed discussion with the authors and found substantial value in this paper. The problem it tackles and the research direction it explores are both exciting and timely.

In my view, the main limitations are that the results are confined to the asymptotic regime and there is no evaluation on real data. However, these factors are not significant enough to ignore the merit of the work. Asymptotic results often provide key insights into the finite-sample regime, and given that the paper positions itself as a theoretical study, the absence of real data experiments is understandable.

Overall, as I mentioned before, I am satisfied with the results presented in the paper and have decided to increase my score to 5. The authors should revise their paper according to our discussion to better reflect its value.

**Limitations:**

The results apply only to the asymptotic setting.

**Quality:**

3

**Strengths And Weaknesses:**

Strengths:

In terms of writing and clarity, this is the most well-written manuscript in my batch. The paper is well-organized, with a smooth flow that clearly articulates both the motivation and the contributions of the work. The problem of adversarial robustness in regression models—both parametric and nonparametric—remains underexplored in the machine learning community, as most existing studies focus on classification. From this perspective, I find the core idea of the paper both timely and compelling.

Weaknesses:

Some of the assumptions in the paper are not well-justified—for instance, the choice of modeling the function as belonging to a second-order Sobolev space. It is unclear whether this choice is made for analytical convenience or if there is a more principled reason behind it. Why not consider a Hölder space instead?

The paper includes only limited simulation results to demonstrate the advantages of the proposed approach, and no experiments have been conducted on real-world datasets. It remains unclear in which practical scenarios this method would be preferable to parametric alternatives. Including empirical insights or intuition based on real data would significantly strengthen the case for its applicability.

The results apply only to the asymptotic setting and consider contamination in the response variable y, but not in the input data x. This is notable since often adversarial certification methods in the literature primarily focus on input perturbations.

---

> ### Author Rebuttal · Authors · 2025-07-31
>
> We thank the reviewer for the helpful feedback. Below we address the questions and will update the manuscript accordingly.
>
> 1. >Some of the assumptions ... Hölder space instead?
>
> The **second-order Sobolev space** is a widely studied and important functional class in nonparametric regression [2].
> Regarding the Hölder space, as mentioned in the paper, there are existing works that have considered this functional setting. Specifically, Zhao et al. analyzed the robustness of an M-estimator based on the Nadaraya–Watson method under a poisoning attack model, assuming the regression function belongs to a Hölder class. Their approach improves robustness, but at the cost of increased computational complexity, specifically $\mathcal{O}(n \log(1/\epsilon))$. This implies that achieving higher accuracy comes with a corresponding increase in computational cost. Another paper studied a different adversarial model, namely, feature corruption, also within the Hölder space framework [1].
> In this paper, we chose to evaluate robustness within the Sobolev setting to complement the existing literature on Hölder spaces. Moreover, the smoothing spline estimator we study admits a closed-form solution and achieves $\mathcal{O}(n)$ computational complexity. It does not involve a trade-off between accuracy and runtime.
>
> 2. >The results apply...input perturbations.
>
> Regarding the **asymptotic setting**, we note that the adversary enjoys substantial flexibility, which makes the theoretical analysis of the problem highly complex. Establishing asymptotic upper and lower bounds is therefore an important step toward understanding the more demanding **non-asymptotic** scenarios.  Even in this  asymptotic regime, the nonparametric nature of the problem poses significant analytical challenges.
>
> In many problems, non-asymptotic bounds remain elusive due to technical obstacles, making it common practice to first develop a solid asymptotic analysis as a foundation.
>
> Regarding the focus on **response (label) corruption**, we note that adversarial learning typically considers two main attack types:
>
> - **Label corruption (poisoning):** where the adversary perturbs the labels $\{y_i\}$ while keeping inputs $\{x_i\}$ fixed.
> - **Input corruption:** where the adversary perturbs the inputs $\{x_i\}$.
>
> Both settings are important in practice and need to be addressed. In this work, we focus on the label corruption model, where up to $q$ labels may be arbitrarily corrupted. This already captures substantial complexity, particularly in the **nonparametric** setting where there are no structural constraints on the regression function.
>
> 3. >One high-level ... adversarial attacks.
>
> As discussed in the paper, regression methods generally fall into two broad categories: **parametric** and **nonparametric**. Both have wide-ranging applications across statistics and machine learning.
>
> Parametric models assume a fixed functional form and have been extensively studied in the context of adversarial robustness. Several such works are cited in our paper. Their structured nature often makes them more amenable to theoretical and algorithmic analysis.
>
> In contrast, **nonparametric regression** does not assume any specific functional form, offering greater flexibility but also introducing significant analytical challenges, particularly under adversarial corruption. Because of these difficulties, far fewer works have studied adversarial robustness in the nonparametric setting.
>
> 4. >While the authors... would be beneficial.
>
> Our setup considers a **much stronger and more general adversarial model**. Specifically, we allow the adversary to **arbitrarily modify up to $q$ labels** $\{y_i\}$, provided the corrupted values lie within a bounded range $[-M, M]$. This setup does **not restrict** the adversary to additive noise, it may employ arbitrary, input-dependent, and even coordinated corruption strategies, potentially tailored to the structure of the estimator.
>
> Regarding the **breakdown point**, our analysis provides a precise characterization for the smoothing spline estimator. We prove that as long as the number of corrupted points satisfies $q = o(n)$, the smoothing spline estimator achieves vanishing error for any function in the second-order Sobolev space. Conversely, when $q/n$ is bounded away from zero, **no estimator** can guarantee vanishing error uniformly over this function class.
>
>
> Finally, while the **median estimator** is known to have a 50% breakdown point in classical settings, this guarantee applies to **point estimation of fixed parameters** (e.g., estimating the mean under i.i.d. noise). Our problem, in contrast, involves **nonparametric function estimation** under adversarial sample corruption, a fundamentally more challenging setting, where the 50% breakdown point does **not** hold.
>
> 5. > I do not understand ...fitted values.
>
> While it is true that the regularization in the smoothing spline objective promotes smoothness and may intuitively mitigate the influence of outliers, the **robustness of smoothing splines is not guaranteed a priori**. In fact, other classical and computationally efficient nonparametric estimators, such as the NW estimator (that implicitly induces some level of smoothness through its kernel and bandwidth) , are **not** robust under adversarial contamination [1]. This highlights that robustness is not an automatic consequence of imposing smoothness.
>
> What makes the smoothing spline particularly surprising and noteworthy is that it is not only **robust**, but also **optimal**., as mentioed in the above question.
>
> 6. >What does $q = \theta(n^\beta)$ mean?
>
> It means there exist constants $c_1, c_2 > 0$ such that for large $n$, $c_1 n^\beta \leq q \leq c_2 n^\beta$.
>
> 7. >The authors discuss... such cases?
>
> Please see the response to question 2.
>
> 8. >In Figure 3 ...results are.
>
> We would like to emphesize that the "best possible estimator" in this settings is not known. In fact, one of the main contributions of our work is to investigate both the **achievability** and the **fundamental limits** of estimation under adversarial corruption.
>
> If, however, the reviewer was referring to including the **theoretical upper bound from Theorem 1** in all experimental plots to better visualize the gap between empirical and theoretical performance, we agree this would be helpful. We will incorporate this in the final version to clarify how closely the smoothing spline tracks the theoretical limits.
>
> 9. >The paper ... applicability.
>
> First, we have extended our experiments as suggested. The detailed results can be found in our response to the following question.
>
> Second, we encourage the reviewer to view this work as addressing a **fundamental and core problem** in robust nonparametric estimation—one that lays the groundwork for robustness in more practical and complex settings.
>
> In particular, smoothing splines under fixed design have recently gained attention in modern machine learning architectures, including **Kolmogorov–Arnold Networks (KANs)** [3], **SplineCNN** [4], and related models [5]. These architectures are being actively used in various ML applications. Our theoretical framework and analysis offer a foundation for establishing **robustness guarantees** in such models, and can potentially guide the design of robust estimators in these and other neural network-based systems.
>
> 10. >In the experiments ... truly are.
>
> We have extended our experiments along several settings to address this concern.
>
> ### Effect of Large $M$
>
> Let $f(x) = x \sin(x)$ over $x \in [-1, 1]$, $M = 10$ and $q = n^{0.6}$.
> $\mathcal{R}_2(f,\hat{f})$
> |Corruption Type|log-log Slope|
> |-|-|
> |Random Corruption|-0.85|
> |Concentrated Corruption|-0.67|
>
> $\mathcal{R}_\infty(f,\hat{f})$
> |Corruption Type|log-log Slope|
> |-|-|
> |Random Corruption|-0.59|
> |Concentrated Corruption|-0.48|
>
> ### Adaptive Greedy Corruption
> In this process, the attacker:
> 1. Fits the baseline estimator on clean data.
> 2. Computes $\ell_i = (\hat{f}(x_i) - y_i)^2$ for each sample: .
> 3. Identifies $i^\star = \arg\min_i \ell_i$.
> 4. Update $y_{i^\star} \leftarrow y_{i^\star} + M \cdot \operatorname{sign}(\hat{f}(x_{i^\star}) - y_{i^\star})$.
> 5. Repeats the process until $q$ points are corrupted.
>
> Below are the results for $f(x) = x \sin(x)$ with $M = 100$ and input domain $[-10, 10]$:
> **$\mathcal{R}_2(f,\hat{f})$ (log-log slope):**
> |$q$|Slope|
> |-|-|
> |$q = n^{0.6}$|-1.12|
> |$q = n^{0.8}$|-0.91|
>
> **$\mathcal{R}_\infty(f,\hat{f})$ (log-log slope):**
> |$q$|Slope|
> |-|-|
> |$q = n^{0.6}$|-1.21|
> |$q = n^{0.8}$|-0.97|
>
> ### **3. Evaluation on Neural Network Functions**
>
> We further tested our approach on a fixed 3-layer MLP neural network and $M = 500$, $\sigma^2 = 1$, and $x \in [-1, 1]$.
> $\mathcal{R}_2(f,\hat{f})$
> |Corruption Type|log-log Slope|
> |-|-|
> |Random Corruption|-0.80|
> |Concentrated Corruption|-0.65|
> |Adaptive Greedy Corruption|-1.29|
>
> $\mathcal{R}_\infty(f,\hat{f})$:
>
> |Corruption Type|log-log Slope|
> |-|-|
> |Random Corruption|-0.86|
> |Concentrated Corruption|-0.48|
> |Adaptive Greedy Corruption|-1.31|
>
> [1] Jingfu *et al.* *Adversarial learning for nonparametric regression: Minimax rate and adaptive estimation*, 2025
> [2] Tsybakov *et al.*, *Introduction to Nonparametric Estimation*, 2008
> [3] Liu *et al.*, *Kan: Kolmogorov-arnold networks.* ICLR 2025
> [4] Fey *et al.*, *Splinecnn: Fast geometric deep learning with continuous b-spline kernels.* CVPR 2018.
> [5] Doległo *et al.*, *Deep neural networks for smooth approximation of physics with higher order and continuity B-spline base functions.* (2022).

---

> > ### Comment · Reviewer_pg9c · 2025-08-01
> >
> > I have carefully read the rebuttal and would like to thank the authors for their hard work in addressing the remaining questions. While I appreciate most of the responses, I remain unconvinced by some of the replies to my questions:
> >
> > **Choice of nonparametric regression**
> >
> > In response to my question about the choice of nonparametric regression, the authors stated, ‘Parametric models assume a fixed functional form and have been extensively studied in the context of adversarial robustness.’ It appears that the main motivation for evaluating the investigated scenario is that it has not been explored before, possibly for the sake of publication. I still believe that using parametric models with a large number of parameters (e.g., large-scale deep networks) may be a more reasonable choice. However, the authors did not provide a convincing argument for why one should prefer a nonparametric model, despite its added complexity. Regarding the imposed structure in parametric settings, I do not believe that such structures—especially those with the degrees of freedom found in deep networks—are restrictive. This point could have been made clearer with real-world applications or a well-designed synthetic example, both of which are currently missing.
> >
> > **Asymptotic results**
> >
> > While I understand the importance of asymptotic analysis in studying the finite-sample behavior of estimators, I still believe that for adversarial analysis and its application to real-world scenarios, the latter is more relevant. I am not opposed to this paper, and I see value in it, as reflected in my score. However, with such insights included, this paper could become significantly stronger and stand a better chance of gaining recognition and credibility within the community.
> >
> >
> > **Differences with standard robust regression framework**
> >
> > Please note that even in standard robust regression models with additive noise, an attacker can arbitrarily modify up to q samples, since the density function h is unknown to the user and can take any form (something similar to Adaptive Greedy Corruption that you have added as the new result)

---

> > > ### Author Response · Authors · 2025-08-05
> > >
> > > We sincerely thank the reviewer for their thoughtful follow-up and for recognizing the value of our work. We appreciate the opportunity to further clarify our motivations and contributions.
> > >
> > > ---
> > >
> > > ### 1. On the Choice of Nonparametric Regression
> > >
> > > While it is true that parametric models—from classical regression to large-scale deep networks—are dominant in practice and have been extensively studied in the context of adversarial robustness, our motivation for studying nonparametric regression goes beyond simply addressing a gap in the literature.
> > >
> > > Nonparametric tools such as splines and kernel-based methods are increasingly being used as building blocks in modern parametric machine learning models [1,2,3], or have been theoretically connected to them [4,5]. A notable example is the Kolmogorov–Arnold Networks (KAN) [1], which incorporate spline interpolation as a core nonlinear component. Studying and understanding the robustness properties of these nonparametric modules is therefore important not only in isolation, but also as a foundation for analyzing the robustness of recent practical deep learning architectures—and potentially improving their robustness through such insights.
> > >
> > > Our results provide provable robustness guarantees for nonparametric regression, and we view this work as a foundation for developing both theoretical and practical insights into the robustness of broader model classes—especially those that integrate nonparametric subcomponents.
> > >
> > > References:
> > >
> > > [1] Liu et al., *KAN: Kolmogorov–Arnold Networks*, ICLR 2025
> > > [2] Hung et al., *Deep P-Spline: Theory, Fast Tuning, and Application*, 2025
> > > [3] Fey et al., *SplineCNN: Fast Geometric Deep Learning with Continuous B-Spline Kernels*, CVPR 2018
> > > [4] Balestriero et al., *A Spline Theory of Deep Learning*, ICML 2018
> > > [5] Jacot et al., *Neural Tangent Kernel: Convergence and Generalization in Neural Networks*, NeurIPS 2018
> > >
> > > ---
> > >
> > > ### 2. On the Relevance of Asymptotic Results
> > >
> > > We truly appreciate the reviewer’s recognition of the value in our theoretical contributions.
> > >
> > > We fully agree that bridging the gap to finite-sample regimes is important, especially for practical adversarial analysis. This paper should be viewed as a foundational step toward broader analyses, including finite-sample and small-data regimes. We are actively working on these directions and plan to incorporate such results in future work.
> > >
> > > ---
> > > ### 3. On Differences with Standard Robust Regression Frameworks
> > >
> > > In standard robust regression, an attacker may arbitrarily corrupt up to $q$ samples. However, this still represents a subset of the strategies allowed in our adversarial model.
> > >
> > > For example, in the additive noise model, as mentioned by the reviewer, a corrupted label is typically modeled as:
> > >
> > > $$
> > > y_i = f(x_i) + e_i,
> > > $$
> > >
> > > where $e_i = (1 - ε)\mathcal{N}(0, σ²) + εh$ for some unknown distribution $h$. While the distribution $h$ may be arbitrary, the corrupted values are still statistically tied to $f(x_i)$—in particular, the expected value of $y_i$ is $f(x_i)$ for clean samples, and $f(x_i) + ε E[h]$ for corrupted ones.
> > >
> > > In contrast, in our adversarial setting, the attacker can assign arbitrary values to $y_i$, possibly entirely independent of the function $f$. For instance, they may select $q$ points and replace their labels with values drawn from a distribution unrelated to $f$, or even specifically chosen to mislead the estimator. This results in a fundamentally **stronger and more challenging** corruption model than what is typically assumed in standard robust regression frameworks.

---

### Official Review · Reviewer_Az91 · 2025-06-12

**Clarity:** 3
**Significance:** 3
**Originality:** 3
**Rating:** 5
**Confidence:** 1

**Summary:**

This work investigates the adversarial robustness of nonparametric regression, focusing on scenarios where a subset of the training labels may be corrupted by adversarial label noise—i.e., an attacker can arbitrarily alter the labels of some training examples. The authors show that, for bounded and one-dimensional input and output spaces, the smoothing spline estimator exhibits robustness to such attacks. Furthermore, they derive a minimax lower bound, establishing that the smoothing spline estimator is statistically optimal under these conditions.

**Questions:**

1. Can the theorem be extended to the case of multiple regression?
2. Can the analysis be extended to classification and feature corruption?

**Ethical Concerns:**

["NO or VERY MINOR ethics concerns only"]

**Limitations:**

Yes.

**Paper Formatting Concerns:**

Do not include acknowledgments at submission time.

**Quality:**

3

**Strengths And Weaknesses:**

Strengths
1. This paper is well-written and easy to follow.
2. The authors propose a minimax lower-bound, which shows the fundamental limits of estimation accuracy.
3. The authors analyze the classical smoothing spline estimator and show the optimal of it.

Weaknesses
1. The input is one-dimensional, which is uncommon in practical applications.
2. The analysis is limited to regression and label corruption.

---

> ### Author Rebuttal · Authors · 2025-07-31
>
> We thank the reviewer for the helpful feedback. Below we address the questions and will update the manuscript accordingly.
>
> > The input is one-dimensional, which is uncommon in practical applications.
>
> **Answer:**
> We thank the reviewer for this comment. Indeed, our analysis focuses on the one-dimensional input setting as an initial step toward understanding adversarial robustness in nonparametric regression.
>
> This setting is already quite challenging: unlike in parametric models, we make no structural assumptions about the underlying function class, and the adversary is allowed to follow an arbitrary contamination strategy. Moreover, as discussed in the paper, not all classical nonparametric estimators are robust in this setting, for instance, the Nadaraya–Watson estimator can be easily misled by adversarial perturbations.
>
> We view our work as a foundational step, and we hope it lays the groundwork for extending these robustness guarantees to higher-dimensional and more complex settings in future work.
>
>
> > The analysis is limited to regression and label corruption.
>
> **Answer:**
> We thank the reviewer for this observation. In adversarial learning, two widely studied forms of attack are:
>
> - **Label corruption (poisoning):** the adversary modifies the labels $\{y_i\}_{i=1}^n$ while keeping the inputs $\{x_i\}_{i=1}^n$ fixed.
> - **Input corruption:** the adversary perturbs the input features $\{x_i\}_{i=1}^n$, possibly relocating them to adversarial positions.
>
> Both types of corruption are important and relevant in practice. In this work, we focus on the **label corruption** setting—formally, we assume the adversary can arbitrarily change the values of $y_i$ for up to $q$ indices.
>
> This setting already captures significant challenges, especially in the **nonparametric** regime where the function class is highly flexible and no structural assumptions are made.  As shown in [Reference], even well-known classical estimators like the Nadaraya–Watson method fail to be robust under this model.
>
> We hope this work lays the foundation for analyzing other adversarial models, including those involving corrupted inputs, in future research.
>
>
> > Can the theorem be extended to the case of multiple regression?
>
> **Answer:**
> We thank the reviewer for this question. While we are not entirely certain what is meant by "multiple regression" in this context, we assume it refers to the setting where the response variable $y$ is a **vector-valued output** of dimension $k$.
>
> The answer is **yes**. Our analysis can be extended to this setting. In both the achievable direction, this is equivalent to applying $ k $ independent smoothing splines (one for each output dimension), all using the same regularization parameter $\lambda$. The convergence rates remain the same as in the scalar case, up to a constant factor that depends on $k$. The same thing happens in lower bound derivation as the problem can be decomposed to $k$ independant problems.
>
> > Can the analysis be extended to classification and feature corruption?
>
> **Answer:**
> We thank the reviewer for this thoughtful question. Regarding **feature corruption**, we note (as discussed in earlier responses) that adversarial attacks generally fall into two broad categories:
>
> - **Label corruption (output attack):** where the adversary modifies the labels $\{y_i\}$ while keeping inputs $\{x_i\}$ fixed.
> - **Feature corruption (input attack):** where the adversary perturbs the features $\{x_i\}$.
>
> In this paper, we focus specifically on **label corruption**, which is already quite challenging in the nonparametric setting due to the lack of structure in the function class and the flexibility of the adversary's strategy. Extending our analysis to include **feature corruption** is a natural and important next step and will likely require additional technical tools, particularly to handle the geometric aspects of input perturbation.
>
> As for **classification**, this belongs to a distinct family of problems with its own objectives and complexity measures. While we do reference several robust nonparametric classification works in the related literature section, our focus in this paper is strictly on nonparametric **regression** under adversarial label contamination.
>
> We view this paper as a foundational step toward broader analysis across different settings, including classification and more general corruption models.

---

### Official Review · Reviewer_MnJT · 2025-07-03

**Clarity:** 2
**Significance:** 3
**Originality:** 2
**Rating:** 4
**Confidence:** 3

**Summary:**

This paper studies the classic problem of non-parametric regression: however, it is studied under an adversarial setting.
To understand the set-up imagine n samples (x_i, y_i) generated from some ground truth f:X  \to Y with y_i = f(x_i) + \eps_i where \eps_i's are mean zero constant variance iid variables (we can even think of X = [0,1]). We wish to learn a function \hat{f} (non-parametric) which fits the data. However, there is an adversary who can choose any q points of their liking and corrupt the y-values there arbitrarily. This makes it unclear if the current toolkit for non-parametric regression is sufficient.

There is prior work in the parametric setting of the above adversarial set-up. However, the setting of non-parametric regression makes the problem challenging due to a lack of "structure" on the hypothesis space.

The main results in the paper are (with error rates being define as squared L_2 norm or squared L_\infty norm between f and \hat{f}):
1> Under some assumptions it is proved that the computationally efficient classic smoothing spline estimator retains robustness against adversarial corruption
2> Further, a lower bound on the error rates of the best data dependent estimator is shown, and under some regime of parameters it matches the error rate achieved by the smoothing spline estimator.

The paper also demonstrates experimental performance of the smoothing spline in presence of random corruptions and concentrated corruptions to corroborate the theory.

**Questions:**

1> Abstract: It will be good to give some notion of what the estimation error is here to help the reader. Also, the last statement about the optimality of the smoothing spline is a bit misleading as that is only true in a certain range of parameters: perhaps you want to reword this part.

2> Typo @ line 38: choose -> chooses, for any -> for all

3> The lines from 45-48, and in fact 63-67, 77-79, whole of section 2 could be made more precise. Chiefly, the game that is being played needs to be specified clearly, i.e., devise a data dependent learner \hat{f} that when given any ground truth f and any collection of data points, and any adversarial corruption of q points, does well on expectation over the randomness of the \epsilon's. Please make sure that this is amply clear as the whole set-up is only made clear when one looks at corollary 3, which is deep insider the paper. The descriptions currently are not making the set-up unambiguous.

4> Typo @ Line 47: any f that belongs...but please reword in light of above

5> Lines 124-129: Can we not just say that we are sampling n points from p: seems that we are morally thinking about it that way

6> Line 130: Do you mean |f(x)| < M/2 or something because if f(x) is M then there is a good chance the answer might go above M for y when the noise is added: so, then the adversary can in fact choose a value outside M.

7> Lower-bound: <a> does this lower-bound match all the assumptions of the upper-bounds...please include a short description of this as we are comparing the two bounds and hence it is good to clarify that even under the assumptions of the upper bound the lower bound works. <b> Can you provide a clean hypothesis testing problem here since there might then be scope to improver the lower-bounds
<c> I was looking at Appendin B...can you explain why you need to have the sampling lemma...can't the adversary just set the sample to (0 + \eps) everytime x< r_q. and trivailly match the sample dist of f_1....I guess it is okay to have a randomized adversary.....also why do you need the linear decay in f_2, can we not have it decay sharply near r_q?.....

8> Experiments: More experiments with other functions and other kind of adversaries (perhaps using some kind of learning algorithms to corrupt the data points)....also why f = x sin(x)?

9> Both Upper and Lower Bounds:  good to emphasize a bit more as to what steps in the proof are novel ..

9> General Suggestion:  It'll be great if we obtain from this work a general understanding of what makes regression robust so as to capture more techniques and not just the smoothing spline operator; even better if this can inform practice

**Ethical Concerns:**

["NO or VERY MINOR ethics concerns only"]

**Final Justification:**

The proof follows a natural skeleton, there are possibly non-trivial steps in the details, but they are buried deep within. I have tried to elicit these from the authors in the review. Overall, while the authors study a natural and important problem, there is a lack of general insight (which would have scored the paper higher) so I have retained my score.

**Limitations:**

Yes (I have included some suggestions in the questions section already)

**Paper Formatting Concerns:**

No concerns

**Quality:**

3

**Strengths And Weaknesses:**

Strengths:

1> Studies an interesting topic: non-parametric regression in the presence of an adversary. This should be of sufficient interest to communities thinking about adversarial robustness.
2> Shows that an existing classic technique, i.e., smoothing spline operator, can tolerate a certain number of adversarial corruptions. This is proved by decomposing the error term and analyzing them by nicely collating previous results. It displays originality in the way the results are combined: the skeleton of the proofs can be argued are perhaps natural.
3> Establishes lower-bound on the optimal rate for any data dependent estimator via a reduction to hypothesis testing. Uses a cool sampling lemma. Though it is not fully explained why such a lemma is needed.

Weakness:

1> Is kind of a stylized setting in the sense of assumptions.
2> No general insight for a theoretical point of view to design robust learners.
3> No crucial insight for practice in terms of dealing with adversarial corruptions beyond using smoothing splines.
4> Experiments are somewhat limited, but that said this is a theory focused paper: so, this is not a big weakness
5> The writing in some areas can be made clearer and steps which are new can be highlighted

---

> ### Author Rebuttal · Authors · 2025-07-31
>
> We thank the reviewer for the helpful feedback. Below we address the questions and will update the manuscript accordingly.
> 1. >Is kind of ... assumptions.
>
> Our goal was to adopt a standard formulation grounded in nonparametric regression. Since classical nonparametric regression does not account for adversarial contamination, we introduced minimal extensions to accommodate such behavior within this framework. We have deliberately chosen assumptions that are both general and conventional. Specifically, we assume the regression function lies in a second-order Sobolev space, which is a standard modeling choice in nonparametric regression[1].
>
> 2. > No general ... learners.
>     >No crucial insight ... splines.
>
> We agree that identifying general theoretical principles for designing robust learners is an important long-term objective. However, due to the scarcity of existing results in this area and the flexibility of the problem formulation in nonparametric regression, we are not yet in a position to propose such general rules.  At this stage, it seems we have no choice but to analyze  each estimator individually, taking into account its specific assumptions, structure, and computational complexity. Notably, even well-established classical methods such as the NW estimator are not inherently robust to adversarial contamination [2].
>
> 3. > Experiments  ... weakness
>     > Experiments: ... why f = x sin(x)?
>
> We have extended our experiments along several settings to address this concern.
>
> ### Effect of Large $M$
>
> Let $f(x) = x \sin(x)$ over $x \in [-1, 1]$, $M = 10$ and $q = n^{0.6}$.
> $\mathcal{R}_2(f,\hat{f})$
>
> |Corruption Type|log-log Slope|
> |-|-|
> |Random Corruption|-0.85|
> |Concentrated Corruption|-0.67|
>
> $\mathcal{R}_\infty(f,\hat{f})$
> |Corruption Type|log-log Slope|
> |-|-|
> |Random Corruption|-0.59|
> |Concentrated Corruption|-0.48|
>
> ### Adaptive Greedy Corruption
> In this process, the attacker:
> 1. Fits the baseline estimator on clean data.
> 2. Computes $\ell_i = (\hat{f}(x_i) - y_i)^2$ for each sample: .
> 3. Identifies $i^\star = \arg\min_i \ell_i$.
> 4. Update $y_{i^\star} \leftarrow y_{i^\star} + M \cdot \operatorname{sign}(\hat{f}(x_{i^\star}) - y_{i^\star})$.
> 5. Repeats the process until $q$ points are corrupted.
>
> Below are the results for $f(x) = x \sin(x)$ with $M = 100$ and input domain $[-10, 10]$:
> $\mathcal{R}_2(f,\hat{f})$:
> |$q$|log-log Slope|
> |-|-|
> |$q = n^{0.6}$|-1.12|
> |$q = n^{0.8}$|-0.91|
>
> $\mathcal{R}_\infty(f,\hat{f})$:
> |$q$|log-log Slope|
> |-|-|
> |$q = n^{0.6}$|-1.21|
> |$q = n^{0.8}$|-0.97|
>
> ### Evaluation on Neural Network Functions
>
> Let $f(x)$ is a 3-layer MLP neural network with $M = 500$, $\sigma^2 = 1$, and $x \in [-1, 1]$.
> $\mathcal{R}_2(f,\hat{f})$
> |Corruption Type|log-log Slope|
> |-|-|
> |Random Corruption|-0.80|
> |Concentrated Corruption|-0.65|
> |Adaptive Greedy Corruption|-1.29|
>
> $\mathcal{R}_\infty(f,\hat{f})$:
>
> |Corruption Type|log-log Slope|
> |-|-|
> |Random Corruption|-0.86|
> |Concentrated Corruption|-0.48|
> |Adaptive Greedy Corruption|-1.31|
>
> 4. > The writing ... highlighted
>
> Please see the response to question 1 and 7.
>
> 5. > Abstract: It ... this part.
>
> As we formally show in the paper, smoothing spline is optimal in terms of the maximum number of adversarial corruptions it can tolerate. Specifically, as long as $q/n \to 0$ as $n \to \infty$, the estimator achieves vanishing  error. On the other hand, when $q/n$ is bounded away from zero, we prove that no estimator can guarantee vanishing error for all functions in the second-order Sobolev space. Therefore, in this sense, the smoothing spline achieves the best possible robustness guarantee.
>
> While we characterize both the achievable convergence rate of the smoothing spline and the minimax lower bound, we don't claim that the smoothing spline is optimal in terms of convergence rate.
>
> 6. > Typo @ line 38
>
> We will correct them in the final version.
>
> 7. > The lines from 45-48 ... unambiguous.
>
> Our intention was to define the problem formulation by building on standard definitions from classical nonparametric regression, with appropriate references to existing baselines. However, since classical regression frameworks do not include any notion of adversarial corruption, we extended the setting to include an adversary capable of modifying up to $q$ labels.
> In doing so, we aimed to adopt widely accepted assumptions and evaluation metrics such as $R_2$ and $R_\infty$, which are standard in the literature [1]. While the formulation builds naturally from these foundations, we understand that some notations and definitions may initially appear dense. We try to make it more accessible to readers.
>
> 8. > Typo @ Line 47
>
> We will correct them in the fincal version.
>
> 9. > Lines 124-129
>
> It is not the case. We note that in our work, we adopt the fixed design assumption [1], where the points $\{x_i\}_{i=1}^n$ are deterministic. Still, we assume that these points satisfy a uniform convergence to a limiting distribution $p(x)$, as $n \to \infty$.  This is a standard assumption in the asymptotic analysis of smoothing spline [3].
>
> 10. > Line 130:
>
> Our formulation involves two boundedness assumptions: first, we assume that the regression function is bounded by some constant $m_1$; and second, that the values chosen by the adversary are also bounded, say by $m_2$. The bound on the adversary's input is implied by the first bound, considering the variance of the additive noise in the honest samples. In our analysis, we simply define $M = \max(m_1, m_2)$.
>
> 11. > Lower-bound: ... sharply near r_q?
>
> ###  Upper vs. Lower Bound
> In the **upper bound**, we assume the target function $f \in \mathcal{W}^2(\Omega)$, and we introduce a specific estimator which is smoothing spline.  On the other hand, the **lower bound** is a **minimax impossibility result**, where we assume $f \in \mathcal{W}^2(\Omega)$, and we derive a lower-bound for **any** estimator.  Thus it is a valid lower-bound for the proposed solution.
>
> ### Hypothesis Testing
> The lower bound is derived via a reduction to a **binary hypothesis testing problem**, following the Proposition 5.1 of Ref. [57] in the paper. In particular, we construct two functions, $f_1$ and $f_2$, with a non-zero distance in both the $L_2$ and $L_\infty$ norms, yet are rendered statistically indistinguishable by the adversary’s attack. Specifically, the adversary modifies at most $q$ labels such that the distribution of the observed data under $f_1$ becomes indistinguishable from that under $f_2$. This implies that any estimator must incur a non-vanishing error for these two functions.
> ### Sampling Lemma
> We appreciate the reviewer for suggesting an alternative. We carefully explored this suggestion and made a sincere effort to make it work. However, our attempts fell short when dealing with the technical details.
> ### Function Shape
> In choosing $f_1$ and $f_2$, we are subject to the constraint that both functions must lie in the Sobolev space $\mathcal{W}^2(\Omega)$. We carefully design $f_2$ using a piecewise construction that satisfies the Sobolev conditions. While alternative constructions are certainly possible, the choice presented here is sufficient to establish the desired lower bound.
>
> 12. > Both Upper ... novel
>
> ### Upper Bound
>
> As noted, one of our main contributions is to analyze the adversarial robustness of the smoothing spline. The error of the estimator has two sources: niose added to the honest inputs, and worst-case adversrial samples. To bound the error of smoothing spline estimator, one of our conribution is to isolate these two errros by introducing an **auxiliary "all-honest" scenario**. We then decompose the overall estimation error (Eq.(7) and Eq. (8)) into two terms:
>
> - The first term measures the error of the smoothing spline estimator in the all-honest setting. This is precisely where we leverage existing results from the literature (Eq.(9) and Eq.(10)).
>
> - The second term captures the deviation between the estimator computed on adversarial data versus the one computed on clean data.
> To bound this second term, we face two key challenges. First, the presence of adversarial nodes means the problem no longer fits within the classical smoothing spline framework, so existing theoretical results are not directly applicable. Second, the adversary's strategy space is infinite-dimensional, which significantly complicates the worst-case analysis.
> To overcome these challenges, we rely on a less commonly used kernel-based representation of the smoothing spline estimator, rather than the standard matrix-based formulation typically adopted in the literature. This choice enables us to decompose and bound the deviation term effectively. By leveraging this representation, along with tools such as equivalent kernels, norm inequalities, and Sobolev interpolation, we derive new upper bounds on the impact of adversarial contamination. Identifying and working with this alternative formulation is an important part of our technical development.
> ### Lower Bound
> To prove the lower-bound, we construct two functions $f_1$ and $f_2$ such that the following conditions are satisfied:
>
> - Their distance in $L_2$ and $L_\infty$ norms is nonzero.
> - An adversary corrupts $q$ labels, such that the corrupted data distributions under $f_1$ and $f_2$ become indistinguishable.
>
> As a result, no estimator can determine whether the data were generated from $f_1$ or $f_2$. Designing the pair $(f_1, f_2)$ to simultaneously satisfy the smoothness and norm gap conditions, along with the explicit construction of an adversarial strategy, constitutes the core technical contribution of our lower bound analysis.
>
> 13. > General Suggestion ... practice
>
> We answered this comment in the second question.
>
> [1] Tsybakov *et al.*, *Introduction to Nonparametric Estimation*, 2008
> [2] Zhao *et al.* *Robust nonparametric regression under poisoning attack.* AAAI 2024
> [3] Silverman *et al.* *Spline smoothing: the equivalent variable kernel method." The annals of Statistics*, 1984

---

> > ### Comment · Reviewer_MnJT · 2025-08-06
> > **Response to authors**
> >
> > I thank the authors for their detailed response. I have gone through them and refined my opinion. However, I still feel that getting a general theoretical/experimental insight would have led me to ascribe an even higher score.
> >
> > I have a few follow up questions which the authors may address:
> >
> > 1> "implying the optimality of the smoothing spline in terms of maximum tolerable number of corrupted samples"...I would still suggest making this line more precise in light of your detailed response
> >
> > 2> Response point <7> ...to someone not thoroughly familiar with the area it would bring clarity if you could devote just a few lines to the precise set-up especially regarding the order of quantifiers....I strongly suggest doing this
> >
> > 3> Response point <10>....I don't see the symbols m_1, m_2 anywhere near line 130...please include this discussion if it is missing
> >
> > 4> Binary hypothesis testing: what I meant was can you describe **quantitatively** what the question is that would lead to better bounds?
> >
> > 5> discussion on the lower-bound...I understand that the lower-bound works for any estimator...I had a different confusion, but I think I understand it now...thanks for the explanation anyway
> >
> > 6> thank you attempting to use the alternative sampling lemma...I wish the concrete sampling question could be spelt out more clearly...but I understand this can be difficult to do :)

---

> > > ### Author Response · Authors · 2025-08-08
> > >
> > > Thank you for the thoughtful follow-up and valuable suggestions
> > >
> > > 1. ⁠⁠We will revise this statement in the paper to more precisely reflect the implications of our results
> > > 2. ⁠We will add a brief explanation of the precise setup, especially clarifying the order of quantifiers, to ensure clarity for readers less familiar with this area.
> > > 3. You are absolutely right. Symbols $m_1$ and $m_2$ do not appear  in the current version. We will include this discussion in the revision to clearly explain the process of selecting $M$
> > > 4. Although we are not entirely sure that we understand the question or not, we clarify the role of the hypothesis testing reduction and how it leads to our lower bound:
> > >
> > > Recall that in our construction, we define two candidate functions $f_1$ and $f_2$, which are distinguishable in $L^2$ norm (i.e., $\|\| f\_1 - f_2 \|\|\_{L^2} > 0$) but made statistically indistinguishable via adversarial corruption. Let $\mathbb{P}^{(q)}_{f_1}$ and $\mathbb{P}^{(q)}\_{f_2}$ denote the distributions over the corrupted datasets when the ground truth is $f_1$ or $f_2$, respectively. The adversary selects a fixed set of $q$ indices (e.g., with $x_i \in [0, r_q]$) and modifies the corresponding $y_i$ labels using a randomized strategy (as described in Lemma 6 of the paper) such that:
> > >
> > > $$
> > > \mathbb{P}^{(q)}\_{f_1} = \mathbb{P}^{(q)}\_{f_2}.
> > > $$
> > >
> > > That is, the total variation distance satisfies:
> > >
> > > $$
> > > \mathrm{TV}(\mathbb{P}^{(q)}\_{f_1}, \mathbb{P}^{(q)}\_{f_2}) = 0.
> > > $$
> > >
> > > This guarantees that **no estimator**—no matter how it is constructed—can distinguish between the two hypotheses based on the corrupted data.
> > >
> > > To formalize this, we invoke **Le Cam's two-point method** (the hypothesis testing between two points), which states that for any estimator $\hat{f}$ and any pair $f_1, f_2$,
> > >
> > > $$
> > > \inf_{\hat{f}} \sup_{f \in \{f\_1, f\_2\}} \mathbb{E} \left[ \| \hat{f} - f \|_{L^2}^2 \right]
> > > \geq  \phi(\frac{\delta}{2})\cdot \left( 1 - \mathrm{TV}(\mathbb{P}^{(q)}\_{f_1}, \mathbb{P}^{(q)}\_{f_2}) \right),
> > > $$
> > >
> > > where $\delta$ denotes the distance between two functions. Choosing $\Phi(x) := x^2$ and the $L_2$-norm as the distance measure, and noting that $\mathrm{TV} = 0$ in our setting, yields the following clean lower bound:
> > >
> > > $$
> > > \inf_{\hat{f}} \sup_{f \in \{f\_1, f\_2\}} \mathbb{E} \left[ \| \hat{f} - f \|_{L^2}^2 \right]
> > > \geq \frac{1}{4} \|\| f_1 - f_2 \|\|\_{L^2}^2.
> > > $$
> > >
> > > We hope this addresses the reviewer's concern accurately.
> > >
> > > 6. We appreciate the reviewer’s understanding.

---

### Official Review · Reviewer_Yr9M · 2025-07-06

**Clarity:** 3
**Significance:** 1
**Originality:** 2
**Rating:** 3
**Confidence:** 4

**Summary:**

The paper analyzes the robustness of nonparametric regression models using splines method. They make the second order Sobolev space assumption on the regression function. They also assume only the labels are arbitrarily corrupted for few samples but not the features. Further the authors prove the minimax optimal lower bound. The authors also claim to have zero  estimation error asymptotically if the number of outliers are limited to o(n).

**Questions:**

1) In lie 131, the authors use the threshold M? How is it chosen in practice? It seems M is treated as a constant in Theorem 1 and its proof or the later discussion. It feels like M can be a large and should not be treated as a constant.
2) Isn’t the truncation to [-M,M] in line 131 used by authors similar to trimming methods critiqued in line 248?

**Ethical Concerns:**

["NO or VERY MINOR ethics concerns only"]

**Final Justification:**

I justify my score because I am still not convinced about how the authors are choosing M, which is critical practically and in theoretical bounds also.

Theoretically, the max of $n$ samples will grow as $O(\log(n))$ even for sub-Gaussian random variables. This may be worse for heavy-tailed data. Hence, the value of M will be at least  $O(\log(n))$. The authors seem to be treating it as a constant.

I don't understand the reasoning behind this comment from authors: "What we meant was that $M$ is chosen to be significantly larger than the range of $f$ over the input domain." If I understand correctly, the range of f is not assumed to be known a priori (otherwise it makes the problem trivial). Hence, I am not sure how they choose the value of M in experiments either. And how do they quantify "significantly larger" practically? I believe these are critical questions in robust statistics.

**Limitations:**

Yes

**Quality:**

2

**Strengths And Weaknesses:**

## Strengths
1) The minimax optimal lower bound seems novel.
2) The derived theoretical results makes sense. For example, achieving zero error for o(n) outliers and impossibility result for constant outlier proportion.
3) The authors have done empirical studies to validate the theoretical claims.

## Weakness

1) Related works should be discussed more. It will help the reader to identify the exact contributions made in the paper
2) In line 284, the authors discard median of means approach saying that it can fail arbitrarily if a single corrupted sample falls in each group. Can the same thing happen in the splines approach?
3) In line 130, the authors the function is bounded in the domain. Does that assumption for simple least squares objective function?
4) There is a clear model mismatch in line 40 and line 144 or 161. It makes it tough to follow the proof in Eq 12
5) So the bounds in Eq 9 and 10 are directly taken from the existing literature? And the bounds for 2nd term in Eq 7 and 8 are not seen in the literature? I am interested in identifying the exact theoretical contributions of this work.
6) The experiments seem to be misleading. In line 215, the authors consider the domain [-10,10] and function f(x) = x sin(x). Hence, the max value of the function can take is 10. The authors take M = 10 in the experiments which basically implies that all outliers greater than 10 were deleted due to truncation or trimming. Ideally, one would not know the exact value of M in practice.

---

> ### Author Rebuttal · Authors · 2025-07-31
>
> We sincerely thank the reviewer for their thoughtful and encouraging feedback.  Here, we provide responses to the questions raised. We will update the final version of the manuscript accordingly.
> 1. > Related works ... the paper.
> ### Our Contribution
> Unlike parametric regression, where robustness has been extensively studied, the nonparametric setting remains more challenging and less explored due to the lack of structural assumptions. To the best of our knowledge, our work is among the few that address adversarial robustness in nonparametric regression.
> A key contribution of our paper is threefold: (1) we demonstrate that the classical smoothing spline estimator, known for its high efficiency and growing popularity [1], is robust to adversarial contamination, and we characterize its convergence rate; (2) we derive minimax lower bounds for this general setting; and (3) by comparing the two, we show that the smoothing spline estimator is optimal in terms of *the maximum number of tolerable outliers*, provided the regularization parameter is appropriately chosen. As noted in the paper and discussed below, this robustness is not necessarily shared by other classical nonparametric methods.
> ### Comparison to The class of NW estimators:
> It has been shown that the NW estimator lacks robustness, even in the presence of a single corrupted sample [2, 3]. As discussed in the paper, concatenating the NW estimator with the **Median-of-Means (MoM)** procedure does not resolve this issue [4].
> Concatenation of the NW estimator with the **Trimmed Mean Estimator** is also not robust against adversarial attacks [3]. In this method, given data $\{(x_i, y_i)\}_{i=1}^n$, the $y_i$ values are sorted, and an $\alpha$ fraction from both ends is removed to eliminate potential outliers. The NW estimator is then applied only to the remaining samples.
> This method is effective under scattered or uniformly distributed outliers, assuming that adversarial samples appear among both high and low extremes. For this reason, it trims data symmetrically from both ends.
> However, it fails under **concentrated attacks**. If the adversary targets a small region of the input space, many honest points may be removed during trimming, while corrupted points remain. This allows the adversary to inject a non-vanishing error into the estimator.
> Compared to the above methods, a key advantage of our approach is that we explicitly analyze the **worst-case adversary**, without assuming uniform contamination. In contrast, prior methods often rely on (implicit) assumptions of near-uniform corruption strategies.
> ### Comparison with Kernel-Based M-Estimators:
> Zhao et al. [2] propose an M-estimator based on the NW method, assuming the regression function lies in a Hölder class. Their approach improves robustness, but at the cost of increased computational complexity, specifically $\mathcal{O}(n \log(1/\epsilon))$. This implies that achieving higher accuracy comes with a corresponding increase in computational cost.
> In contrast, our method operates over the second-order Sobolev space, which is a different functional class. Moreover, the smoothing spline estimator we study admits a closed-form solution and achieves $\mathcal{O}(n)$ computational complexity. It does not involve a trade-off between accuracy and runtime.
>
> Other works have studied adversarial robustness in nonparametric settings such as classification and density estimation, as covered in the paper.
>
> 2. > In line 284 ... splines approach?
>
> No, this issue does not occur in the spline approach.  The smoothing spline fits a global function with regularization. This smoothness constraint limits the influence of individual outliers.  As proven in the paper, the smoothing spline can tolerate up to $o(n)$ adversarial samples. In addition, we have proved that if $q/n$ stays bounded away from zero, then no estimator can achieve vanishing error for all functions in the Sobolev space.
>
> 3. >In line 130 ... function?
>
> We were unable to fully understand the second part of the question, as it appears that some words were accidentally dropped. Still, we would like to emphasize that the assumption of a bounded function over the domain is a standard practice in robustness analysis for nonparametric regression [2, 3]. The metrics we use, $R_2$ and $R_\infty$, are also well-established in the literature [5].
> Our formulation involves two boundedness assumptions: first, we assume that the regression function is bounded by some constant $m_1$; and second, that the values chosen by the adversary are also bounded, say by $m_2$. The bound on the adversary's input is implied by the first bound—namely, that the adversary cannot inject arbitrarily large inputs. In our analysis, we simply define $M = \max(m_1, m_2)$.
> Importantly, $M$ is not a parameter used by the algorithm itself. It does not influence the computation or tuning of the estimator. Any sufficiently loose upper bound suffices, as long as it is finite. Varying $M$ affects only the constants in the theoretical bounds and not the asymptotic convergence rates.
>
> 4. > There is a clear ... in Eq 12.
>
> Note that in the original setting, a subset of nodes is honest, while the rest are adversarial. To isolate and evaluate the estimator's deviation caused by adversarial contamination, from the deviation caused by the added noise, we define a hypothetical scenario in which all nodes, including those previously adversarial, are assumed to be honest and follow the same noise model. We refer to this as the *all-honest* scenario.
> In particular, the labels in the hypothetical *all-honest* scenario, denoted by $y_i$, match the labels in the adversarial setting, $\tilde{y}_i$, for all $i \in [n] \setminus \mathcal{A}$. For the remaining points where $i \in \mathcal{A}$, we define $y_i = f(x_i) + \epsilon_i$, where $\epsilon_i$ follows the standard noise distribution.
> The key difference is that in this auxiliary setting, the honest nodes remain unchanged, while the adversarial nodes are hypothetically replaced with honest ones that generate data according to the assumed statistical model. To reflect this, we introduce a new set of noise variables for these now-honest nodes, denoted by $\hat{\epsilon}$ in line 145.
> We will revise the relevant section of the manuscript to clarify this modeling step and explain the motivation and notation, so the distinction is clear to the reader.
>
> 5. >So the bounds in Eq 9 ... this work.
>
> As noted, one of our main contributions is to analyze the adversarial robustness of the smoothing spline.  The error of the estimator has two sources: niose added to the honest inputs, and worst-case adversrial samples. One of our conribution is to isolate these two errros by introducing an **auxiliary "all-honest" scenario**. We then decompose the overall estimation error (as shown in Eq. (7) and Eq. (8)) into two terms:
> - The first term measures the error of the smoothing spline estimator in the all-honest setting. This is where we leverage existing results from the literature (Eq.(9) and Eq.(10)).
> - The second term captures the deviation between the estimator computed on adversarial data versus the one computed on clean data. This part lies outside the scope of existing analyses of smoothing splines.
>
> To bound the second term, we face two key challenges. First, the presence of adversarial nodes means the problem no longer fits within the classical smoothing spline framework, so existing theoretical results are not directly applicable. Second, the adversary's strategy space is infinite-dimensional, which significantly complicates the worst-case analysis.
> To overcome these challenges, we rely on a less commonly used **kernel-based representation** of the smoothing spline estimator, rather than the standard matrix-based formulation typically adopted in the literature. This choice enables us to decompose and bound the deviation term effectively. By leveraging this representation, along with tools such as equivalent kernels, norm inequalities, and Sobolev interpolation, we derive new upper bounds on the impact of adversarial contamination. Identifying and working with this alternative formulation is an important part of our technical development.
>
> 6. >The experiments seem to ... value of M in practice.
>
> For the choice of $M$, please see the answer to the question 3. In addition, we have done another experiment for a large $M$ compared to the input domain, as follows (plots are not available due to NeurIPS policy):
> | $M = 10,\ x \in [-1,1]$, $q = n^{0.6}$ |
>
> |$\mathcal{R}_2(f,\hat{f})$|log-log Slope|
> |-|-|
> |Random Corruption|-0.85|
> |Concentrated Corruption|-0.67|
>
> |$\mathcal{R}_\infty(f,\hat{f})$|log-log Slope|
> |-|-|
> |Random Corruption|-0.59|
> |Concentrated Corruption|-0.48|
>
> 7. >In lie 131 ... as a constant.
>
> Plese see the response to the question 3.
>
> 8. >Isn’t the truncation ... line 248?
>
> In trimming-based estimators, the core idea is to eliminate outliers. Consequently these methods are not robust against adversarial attacks, where the labels are strategically shifted within the acceptable range to induce an uncompensated error. As a result, this approach is NOT robust against worst-case adverserial attack (see answer to question 1).
> In contrast, the core idea of smoothing splines is to enforce smoothness by adding a regularization term, scaled by an appropriate factor. This imposed smoothness counterbalances adversarial shifts, even when those shifts lie within the acceptable range. This effect is implied by Theorem 1.
>
> [1] Wahba *et al.*, *Smoothing noisy data with spline functions.* 1975.
> [2] Zhao *et al.*, *Robust nonparametric regression under poisoning attack*, AAAI 2024.
> [3] Dhar *et al.* *The trimmed mean in non-parametric regression function estimation*, 2022.
> [4] Ben-Hamou, Anna, and Arnaud Guyader. "Robust non-parametric regression via median-of-means." 2023.
> [5] Tsybakov *et al.*, *Introduction to Nonparametric Estimation*, 2008

---

> > ### Author Response · Authors · 2025-08-06
> >
> > Dear Reviewer Yr9M,
> >
> > As we approach the conclusion of the author-reviewer discussion phase, we wish to gently remind you that we remain available to address any additional questions or concerns you may have before finalizing your score.

---

> > ### Comment · Reviewer_Yr9M · 2025-08-06
> >
> > I thank the authors for their detailed response. Unfortunately, some of the fundamental concerns are still unresolved. Hence, I intend to retain my score. The detailed comments are as follows:
> >
> > >  In addition, we have done another experiment for a large $M$ compared to the input domain
> >
> > I do not understand the motivation behind comparing the value of M to input domain. As per line 99, the authors are having outliers in response variable y and not features x. So ideally the value of M should be taken depending on the range and not domain.
> >
> > > In line 131, the authors say Extremely large outlier values are excluded, as they can be easily detected and removed.
> >
> > The authors assume that all large values are outliers. This is not true for heavy-tailed data.
> >
> > Also, all the adversarial assumptions should be stated clearly in the beginning (like in the equation after line 99 instead of line 131). The authors have mentioned * for entries in $\mathcal{A}$ in the equation after 99. This makes the reader think that the adversary can choose the values arbitrarily.
> >
> > > No, this issue does not occur in the spline approach. The smoothing spline fits a global function with regularization.
> >
> > The splines approach minimizes the impact of (bounded) outliers, but does it nullify the effect of outliers? I feel outliers can still propagate their effect across a large region due to the global smoothness penalty for large value of M (which is ideally unknown)
> >
> > The authors have criticized the median of means and the trimmed mean estimator. Is this criticism for a general adversary where the adversary can choose any value? Or is it for the restrictive adversary where it can choose values in [-M,M] which they consider in their paper?
> >
> > > Importantly, $M$ is not a parameter used by the algorithm itself. It does not influence the computation or tuning of the estimator.
> >
> > It sounds like the authors are claiming that their algorithm treats a large or small outlier in the same way. If that is correct understanding, can they provide more intuition as to why that is intended? I can understand if a large or small outlier sample is treated the same way. But I think the aftereffects of a large outlier using splines will be worse as compared to a small outlier.

---

> > > ### Author Response · Authors · 2025-08-08
> > >
> > > We sincierly thank the reviwer for additinoal comments.
> > >
> > > > I do not understand the  ...
> > >
> > > This was a writing mistake in the rebuttal, and we apologize if it caused any confusion.
> > >
> > > What we meant was that $M$ is chosen to be significantly larger than the range of **$f$ over the input domain**.
> > >
> > > > The authors  ... for heavy-tailed data.
> > > > Also, all the adversarial... arbitrarily.
> > >
> > >  We thank the reviewer for raising this point and we want to clarify it as follows:
> > >
> > > Our proofs require only the following bounds:
> > >
> > > 1. The target function $f(\cdot)$ is bounded over its domain $\Omega$ by $m_1$, i.e., $|f(x)| \leq m_1$.
> > > 2. The adversary’s corruption level is bounded by $m_2$.
> > > 3. The variance of the data noise $\epsilon$ is bounded by $\sigma^2$.
> > >
> > > By setting $M = \max(m_1, m_2)$, our proofs remain valid. The reviewer is correct that with heavy-tailed noise, some honest data values may exceed $M$. However, as long as the noise variance is bounded, our results hold.
> > >
> > > Therefore, we do not assume that all values are strictly within $[-M, M]$ or treat large values as outliers.
> > >
> > > We will revise lines 130–131 to clarify the process of selecting $M$. Additionally we will consolidate all adversary-related assumptions in a single section to improve clarity and avoid confusion.
> > >
> > >
> > > >The splines approach ... (which is ideally unknown)
> > >
> > > >The authors ... their paper?
> > >
> > > For both points, we would like to note that, as mentioned in the related work section and in more detail in Question 1 of the rebuttal, we discuss two related works that aim to improve the NW estimator's robustness using classical robust statistics: one using median-of-means (MoM) and the other using trimmed mean, which we refer to as MOM-NW and TM-NW, respectively. Our comparisons in the paper are between our proposed method and these two NW-based robust variants.
> > >
> > > Note that the *original* MoM and trimmed mean estimators are designed for robust **mean estimation**, not nonparametric regression, and therefore are **not directly comparable** to our setting.
> > >
> > > **First**, we would like to provide some intuition for why the NW estimator can collapse with a single bounded adversary:
> > >
> > > The NW estimator is defined as
> > > $$
> > > \hat{f}(x) = \frac{\sum\_{i=1}^N K( \frac{x - x\_i}{h} ) y_i}{\sum\_{i=1}^N K( \frac{x - x\_i}{h} )},
> > > $$
> > > where $K(\cdot)$ is a kernel function and $h$ is the bandwidth parameter.
> > >
> > > Suppose $y_1 \in [-M, M]$ is corrupted and $x_1$ is extremely close to the query point $x$, so that $K\left( \frac{x - x_1}{h} \right) \approx 1$, while the remaining $x_i$ are far from $x$, so that $K\left( \frac{x - x_i}{h} \right) \approx 0$ for $i > 1$. Then:
> > > $$
> > > \hat{f}(x) \approx \frac{K(0) \cdot y_1}{K(0)} = y_1
> > > $$
> > > and the estimation error is
> > > $$
> > > |\hat{f}(x) - f(x)| = |y_1 - f(x)|,
> > > $$
> > > which does **not** vanish as $N \to \infty$.
> > >
> > > This shows the NW estimator can be **collapsed by a single outlier**, even when the adversarial value is bounded within $[-M, M]$.
> > >
> > > Thus, the limitations we discussed in the related work section for MOM-NW and TM-NW apply even in the **bounded** adversarial setting.
> > >
> > > **Second**, as the reviewer correctly points out, outliers can inject more error as the bound $M$ increases. However, in the rebuttal, when we stated *"No, this issue does not occur in the spline approach"*, we were specifically addressing the reviewer’s earlier concern:
> > >
> > > > *“The median-of-means approach can fail arbitrarily if a single corrupted sample falls into each group. Can the same thing happen in the spline approach?”*
> > >
> > > Our point was that the spline method does **not** suffer from the single-adversary failure and other limitations seen in NW and its other variants MOM-NW and TM-NW described above. As discussed in the paper, the smoothing spline estimator only fails when $q/N \not \to 0$ as $N \to \infty$.
> > >
> > >
> > >
> > > > It sounds like the ... as compared to a small outlier.
> > >
> > > We would like to clarify our statement. In the last line of our third rebuttal response, we wrote:
> > >
> > > `Varying $M$ affects only the constants in the theoretical bounds and not the asymptotic convergence rates.`
> > >
> > > What we meant is that changing $M$ does **not** affect the **convergence rates** of $R_2$ or $R_\infty$, it appears only as a **constant factor** in the error bounds. While larger $M$ certainly increases the absolute errors, the **rate** at which they decay with more data stays the same.
> > >
> > > ---
> > >
> > > We hope that we were able to address your remaining concerns, and would be happy to clarify any point if needed. "

---

### Decision · Program_Chairs · 2025-09-17

**Decision:**

Accept (poster)

**Comment:**

This paper studies the adversarial robustness of nonparametric regression under contamination, providing novel upper and lower bounds on estimation rates in second-order Sobolev spaces, and demonstrating how smoothing spline estimators achieve minimax-optimal robustness. The work fills a gap in the literature, as most prior robustness analyses have focused on parametric models.

The reviewers found the problem important and the contributions significant, though raised concerns about clarity of assumptions (e.g., $M$ that grows logarithmically with $n$), technical correctness in parts of the proofs, and the positioning of the results relative to prior work. Reviewer comments highlighted presentation issues and asked for clarification of key modeling choices.

The authors provided a detailed and thoughtful rebuttal that addressed these concerns point by point, clarifying assumptions, correcting inconsistencies, and expanding discussion of related literature and implications. While some reservations about exposition remain, the responses and additional experiments strengthened confidence in the correctness and relevance of the results.

Overall, weighing the novelty, technical depth, and potential impact against presentation concerns, I find the paper makes an important and timely contribution to the theory of robust learning. I therefore recommend acceptance.